# Uncontrolled angiogenic precursor expansion causes coronary artery anomalies in mice lacking *Pofut1*

Yidong Wang[1], Bingruo Wu[1], Pengfei Lu[1], Donghong Zhang[1], Brian Wu[1], Shweta Varshney[2], Gonzalo del Monte-Nieto [3,4], Zhenwu Zhuang[5], Rabab Charafeddine[6], Adam H. Kramer[6], Nicolas E. Sibinga [7], Nikolaos G. Frangogiannis[8], Richard N. Kitsis [9], Ralf H. Adams [10], Kari Alitalo[11], David J. Sharp[6], Richard P. Harvey[3,4,12], Pamela Stanley [2] & Bin Zhou [1,13]

Coronary artery anomalies may cause life-threatening cardiac complications; however, developmental mechanisms underpinning coronary artery formation remain ill-defined. Here we identify an angiogenic cell population for coronary artery formation in mice. Regulated by a DLL4/NOTCH1/VEGFA/VEGFR2 signaling axis, these angiogenic cells generate mature coronary arteries. The NOTCH modulator POFUT1 critically regulates this signaling axis. POFUT1 inactivation disrupts signaling events and results in excessive angiogenic cell proliferation and plexus formation, leading to anomalous coronary arteries, myocardial infarction and heart failure. Simultaneous VEGFR2 inactivation fully rescues these defects. These findings show that dysregulated angiogenic precursors link coronary anomalies to ischemic heart disease.

[1] Departments of Genetics, Pediatrics, and Medicine (Cardiology), Wilf Cardiovascular Research Institute, Albert Einstein College of Medicine, Bronx, New York 10461, USA. [2] Department of Cell Biology, Albert Einstein College of Medicine, Bronx, NY 10461, USA. [3] Developmental and Stem Cell Biology Division, Victor Chang Cardiac Research Institute, Darlinghurst, NSW 2010, Australia. [4] St. Vincent's Clinical School, University of New South Wales, Kensington, NSW 2052, Australia. [5] Department of Medicine, Yale University, New Haven, Connecticut 06510, USA. [6] Department of Physiology and Biophysics, Albert Einstein College of Medicine, Bronx, New York 10461, USA. [7] Departments of Medicine, Developmental and Molecular Biology, Wilf Cardiovascular Research Institute, Albert Einstein College of Medicine, Bronx, New York 10461, USA. [8] Departments of Medicine, Microbiology and Immunology, Wilf Cardiovascular Research Institute, Albert Einstein College of Medicine, Bronx, New York 10461, USA. [9] Departments of Medicine and Cell Biology, Wilf Cardiovascular Research Institute, Albert Einstein College of Medicine, Bronx, New York 10461, USA. [10] Department of Tissue Morphogenesis, Max-Planck-Institute for Molecular Biomedicine, Röntgenstraße 20, and Faculty of Medicine, University of Münster, 48149 Münster, Germany. [11] Wihuri Research Institute, Biomedicum Helsinki, Haartmaninkatu 8, FI-00290 Helsinki, Finland. [12] School of Biotechnology and Biomolecular Science, University of New South Wales, Kensington, NSW 2052, Australia. [13] Department of Cardiology, The First Affiliated Hospital of Nanjing Medical University, Nanjing, Jiangsu 210029, China. Correspondence and requests for materials should be addressed to B.Z. (email: bin.zhou@einstein.yu.edu)

Coronary arteries provide nutrients and oxygen to cardiac muscle, and are therefore essential for heart functions[1, 2]. Consisting of three tissue layers (an inner layer of endothelium, middle layer of vascular smooth muscle and outer layer of fibroblasts), coronary arteries are formed by a tightly-regulated and complex process, disruption of which may cause coronary artery anomalies, leading to sudden cardiac death[3, 4], myocardial infarction[5] or heart failure[6]. In mice, development of coronary vessels begins with the formation of an endothelial plexus by precursor cells arising in ventricular endocardium[7], sinus venous endocardium[8], and proepicardium/epicardium[9]. Regardless of their origins, these progenitors invade the myocardium and proliferate to form coronary plexuses (primitive coronary vessels) by vasculogenesis (de novo vessel formation). The deeper part of the coronary plexus is specified to an arterial fate and coalesces then recruits pericytes to become mature coronary arteries with a smooth muscle media[10, 11]. Despite the profound importance of coronary arteries, the characteristics of coronary angiogenic precursor cells and the molecular mechanisms that drive them to form coronary arteries remain poorly understood.

NOTCH signaling regulates multiple cellular functions in heart development and disease[12, 13]. Notch genes encode transmembrane receptors (NOTCH1-4) that interact with membrane-bound ligands of the Delta (DLL1,3,4) and Jagged (JAG1,2) family[14]. The binding of NOTCH ligands is regulated by protein O-fucosyltransferase 1 (POFUT1). Bound ligand triggers cleavage of the associated receptor and then release of its intracellular domain (NOTCH intracellular domain, which shuttles to the nucleus where it forms a transcriptional complex with RBPjk and activates the expression of target genes[15, 16]. NOTCH signaling functions as a key regulator of angiogenesis outside of the heart in physiological and pathological conditions[17–19], in particular arterial-venous specification[20, 21], tip and stalk cell selection during sprouting angiogenesis[22–26], proliferation and survival of endothelial cells[27], and smooth muscle recruitment and differentiation[28–32]. Whether NOTCH signaling plays similar roles in coronary artery development is incompletely understood[31]. This is partly due to lack of functional definition of progenitor cells for coronary arteries.

Here we functionally characterize a coronary angiogenic precursor population, which expresses high levels of VEGFR3. We show that POFUT1, a regulator of NOTCH ligand binding, regulates this precursor pool through DLL4/NOTCH1/VEGF signaling. Disruption of Pofut1 in ventricular endocardium results in structural coronary artery anomalies and early-onset ischemic heart disease due to insufficient coronary oxygen perfusion. Loss of Pofut1 promotes proliferation of angiogenic precursors to form excessive coronary plexuses, which fail to undergo proper arteriogenesis.

## Results

**Coronary artery anomalies in mice lacking endocardial Pofut1.** To investigate functions of POFUT1, an upstream modulator of ligand-induced NOTCH signaling[33–35], in heart development, we deleted Pofut1 in individual cardiac cell lineages by crossing floxed Pofut1 mice with tissue-specific Cre lines including TntCre for myocardium, Tbx18Cre for epicardium, Sm22αCre for vascular smooth muscle, Mef2cCre for secondary heart field tissue, Tie1Cre for vascular endothelium, and Nfatc1Cre for cardiac endocardium. Mice with Pofut1 deletion in myocardium, epicardium, vascular smooth muscle, or secondary heart field were healthy and lived to adulthood (Supplementary Table 1). In contrast, deletion of Pofut1 in endocardium (Nfatc1Cre;Pofut1f/f, hereafter Pofut1cKO) (Supplementary Fig. 1) resulted in death of ~ 80% Pofut1cKO mice by 4 months of age (Fig. 1a). Nearly 40% of endocardial

Pofut1cKO died of heart failure at a young age between postnatal day (P) 40 and P120, indicated by dilated heart (Fig. 1b), increased heart/body weight (Fig. 1c), and impaired cardiac function, including increased left ventricular volume and reduced fractional shortening as well as ejection fraction at P60 (Fig. 1d). There was also elevated Myh7 and decreased Myh6 expression in diseased hearts (Fig. 1e), a 'molecular switch' indicative of heart failure[36]. Pathology evaluations by Hematoxylin/Eosin and Sirius Red staining revealed that diseased hearts had enlarged chambers, sub-endocardial fibrosis, and grossly dilated coronaries at P60 (Supplementary Fig. 2a–e). Another 40% of Pofut1cKO mice died at an even younger stage between P19 and P40 with severe myocardial infarction, as evidenced by large areas of myocardial necrosis and fibrosis (Fig. 1a, f, g, and Supplementary Fig. 2f). Of note, before myocardial infarction, the myocardium of Pofut1cKO mice was already severely hypoxic (Fig. 1h) and some cardiomyocytes underwent apoptosis, whereas no apoptotic cells were found in the myocardium of control hearts (Supplementary Fig. 2g). These observations indicated poor coronary perfusion in Pofut1cKO hearts. Micro-computed tomography confirmed dysfunctional coronary arteries by revealing hypoplastic main coronary arteries and aneurysms (Fig. 1i, Supplementary Fig. 3a). Quantitative analysis of coronary arterial trees showed that Pofut1cKO hearts had a decreased number of middle- to large-size arteries, and an increased number of small arteries (Supplementary Fig. 3b). Antibody staining for PECAM1, a marker of vascular endothelium showed increased coronary plexus formation in Pofut1cKO hearts at P15 (Fig. 1j, Supplementary Fig. 2h). These findings demonstrate that endocardial loss of Pofut1 causes coronary artery anomalies, leading to early-onset ischemic heart disease.

**POFUT1 is required for coronary arteriogenesis.** Coronary structural anomalies have a fetal origin. We therefore examined the embryonic coronary arteries of control and Pofut1cKO hearts using coronary arteriography with fluorescent dye. The results showed that Pofut1cKO hearts had already developed hypoplastic main coronary arteries at embryonic day (E) 16.5 (Fig. 2a). Co-immunostaining with antibodies for PECAM1 and DLL4 (predominantly expressed by arterial endothelium) confirmed that Pofut1cKO hearts had underdeveloped main coronary arteries in the outer myocardium close to the epicardium. In addition, the density of small coronary arteries in the inner myocardium close to the endocardium was significantly increased (Fig. 2b). Co-immunostaining of Isolectin B4 (IB4) expressed in endothelium and SMMHC (smooth muscle myosin heavy chain) or αSMA (alpha-smooth muscle actin) for vascular smooth muscle revealed that Pofut1cKO coronary arteries had poor smooth muscle media (Fig. 2c, Supplementary Fig. 4a). Pofut1cKO hearts at E16.5 also had decreased expression of smooth muscle marker genes (Smmhc, Sm22α and αSMA) and NOTCH pathway genes (Dll4, Jag1, Notch3, Notch4, HeyL and Hes5) (Supplementary Fig. 4b). Additionally, expression of cell junction genes (Itgb3, Cx37 and Cx40) involved in endothelium-pericyte adhesion was decreased in E16.5 Pofut1cKO hearts, whereas expression of genes involved in pericyte differentiation (PDGFB, TIE-Angiopoietin and TGFB pathway genes) or arterial and venous endothelial markers was not affected by deletion of Pofut1 (Supplementary Fig. 5). Consistently, functional assays using E11.5 ventricle explant culture suggested defective attachment of SM22α+ pericytes to the extruding endocardium-derived endothelial cells, which were genetically labeled by GFP using the Nfatc1Cre driver (Fig. 2d). The structural coronary artery anomalies in Pofut1cKO hearts led us to examine coronary perfusion. Hypoxia probe labeling showed that the inner myocardium of E16.5 control hearts was

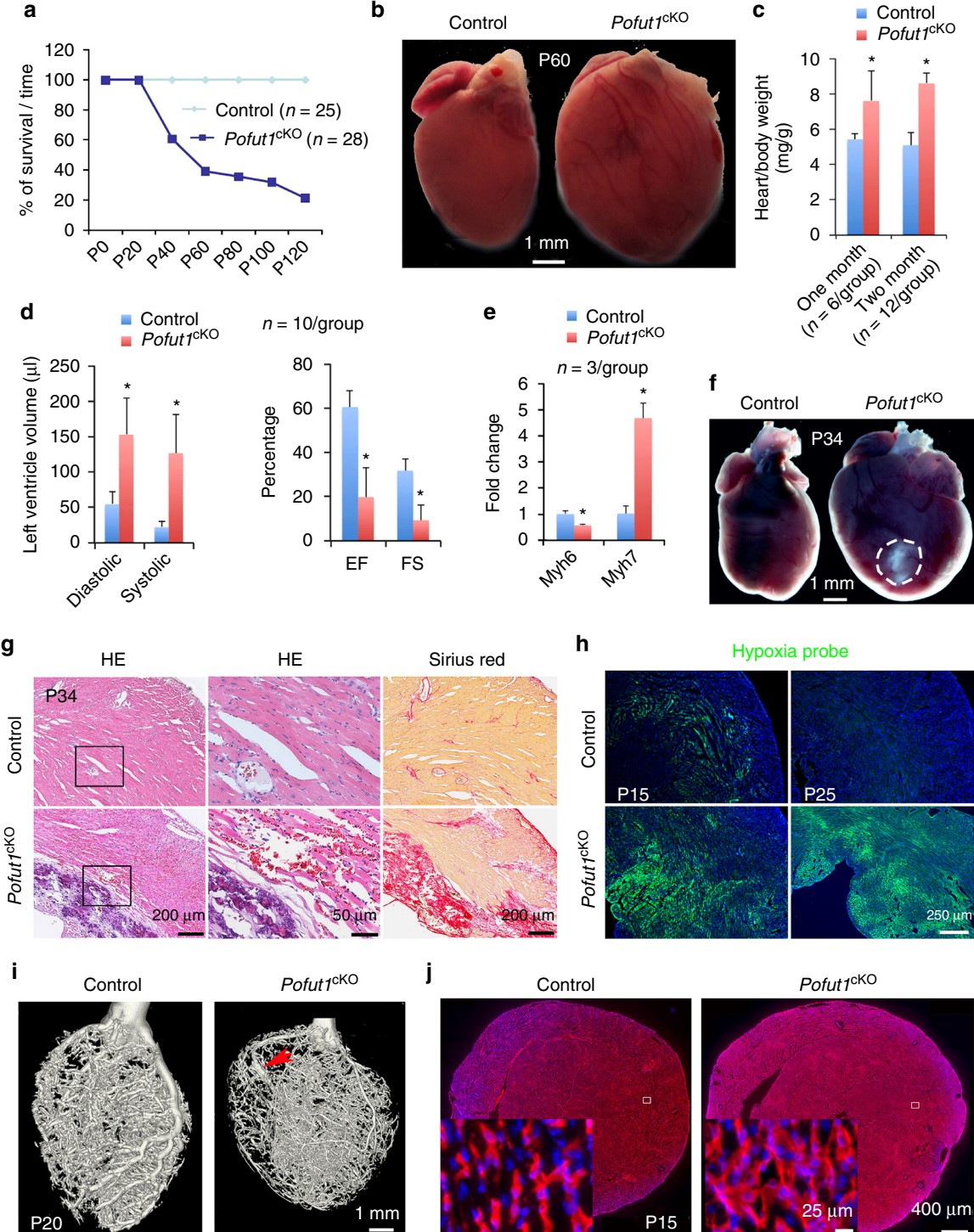

**Fig. 1** Loss of *Pofut1* causes coronary artery anomalies and ischemic heart disease. **a** Survival curve shows ~80% *Nfatc1*[Cre];Pofut1[f/f] (*Pofut1*[cKO]) mice die by P120. **b** Gross view of a dilated P60 *Pofut1*[cKO] heart. **c** Heart/body ratio. *$p < 0.01$. **d** Echocardiography showing reduced cardiac functions in P60 *Pofut1*[cKO] heart. *$p < 0.001$. **e** qPCR showing reversed expression changes of *Myh6* and *Myh7* in P21 *Pofut1*[cKO] hearts. *$p < 0.01$. **f** Gross view of a large myocardial infarct (*circle*) at the apex of a P34 *Pofut1*[cKO] heart. **g** Cardiac histology showing cardiac necrosis and fibrosis in a P34 *Pofut1*[cKO] heart. **h** Hypoxia staining showing expansion of inner myocardial hypoxia in young *Pofut1*[cKO] mice. **i** MicroCT shows coronary anomalies with hypoplastic main arteries and aneurisms (*arrow*). **j** PECAM1 staining shows increased primitive coronary plexuses in a P15 *Pofut1*[cKO] heart. All bar charts represent mean ± s.d

hypoxic and this hypoxic region was largely expanded in *Pofut1*[cKO] hearts (Fig. 2e, f). This result suggests that the increased coronary plexuses in the inner myocardium of *Pofut1*[cKO] hearts are dysfunctional and immature vessels. Consistent with the hypoxic finding, VEGFA antibody staining showed that relatively high levels of VEGFA were located in the inner myocardium of control hearts and this VEGFA-positive region was dramatically expanded in *Pofut1*[cKO] hearts (Fig. 2g, h).

To determine whether endocardial deletion of *Pofut1* affected formation of coronary veins and cardiac lymphatic vessels, we

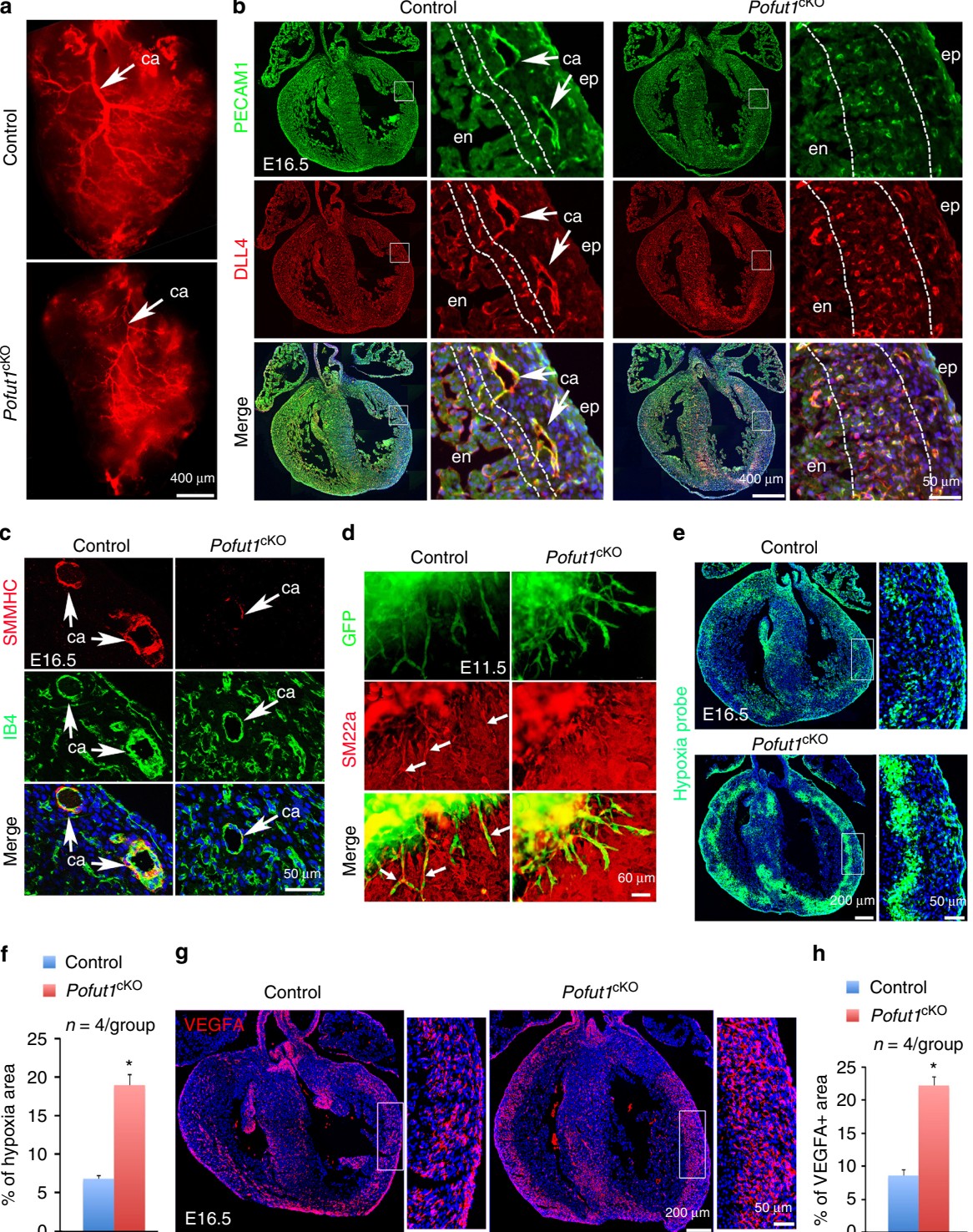

**Fig. 2** POFUT1 is required for coronary artery formation. **a** Coronary arteriography using red fluorescent dye showing underdeveloped coronary arteries (ca, *arrow*) in a E16.5 *Pofut1*cKO heart. (**b**) Co-immunostaining for PECAM1 and DLL4 on E16.5 of control and *Pofut1*cKO heart sections. High magnification images on right showing *boxed* regions in the individual left ventricles with smaller main arteries (*arrows*) and expansion of arterial plexus in the inner myocardium (*between dotted lines*) in a *Pofut1*cKO heart. **c** SMMHC and Isolectin-B4 (IB4) co-immunostaining showing hypoplastic major coronary arteries with thin smooth muscle layer (*arrow*) in a *Pofut1*cKO heart. **d** E11.5 ventricular explant culture (GFP labels ventricular endocardial-derived plexus, red fluorescence labels pericytes using SM22α antibodies) showing poor pericyte attachment to the plexus in a *Pofut1*cKO explant. **e, f** Representative hypoxia probe staining and quantifications show expansion of inner myocardial hypoxic zone in an E16.5 *Pofut1*cKO heart. *$p < 0.01$. **g, h** Representative VEGFA immunostaining and quantifications show expansion of VEGFA expression in the myocardial wall of an E16.5 *Pofut1*cKO heart. *$p < 0.01$. *en/ep* endocardium/epicardium

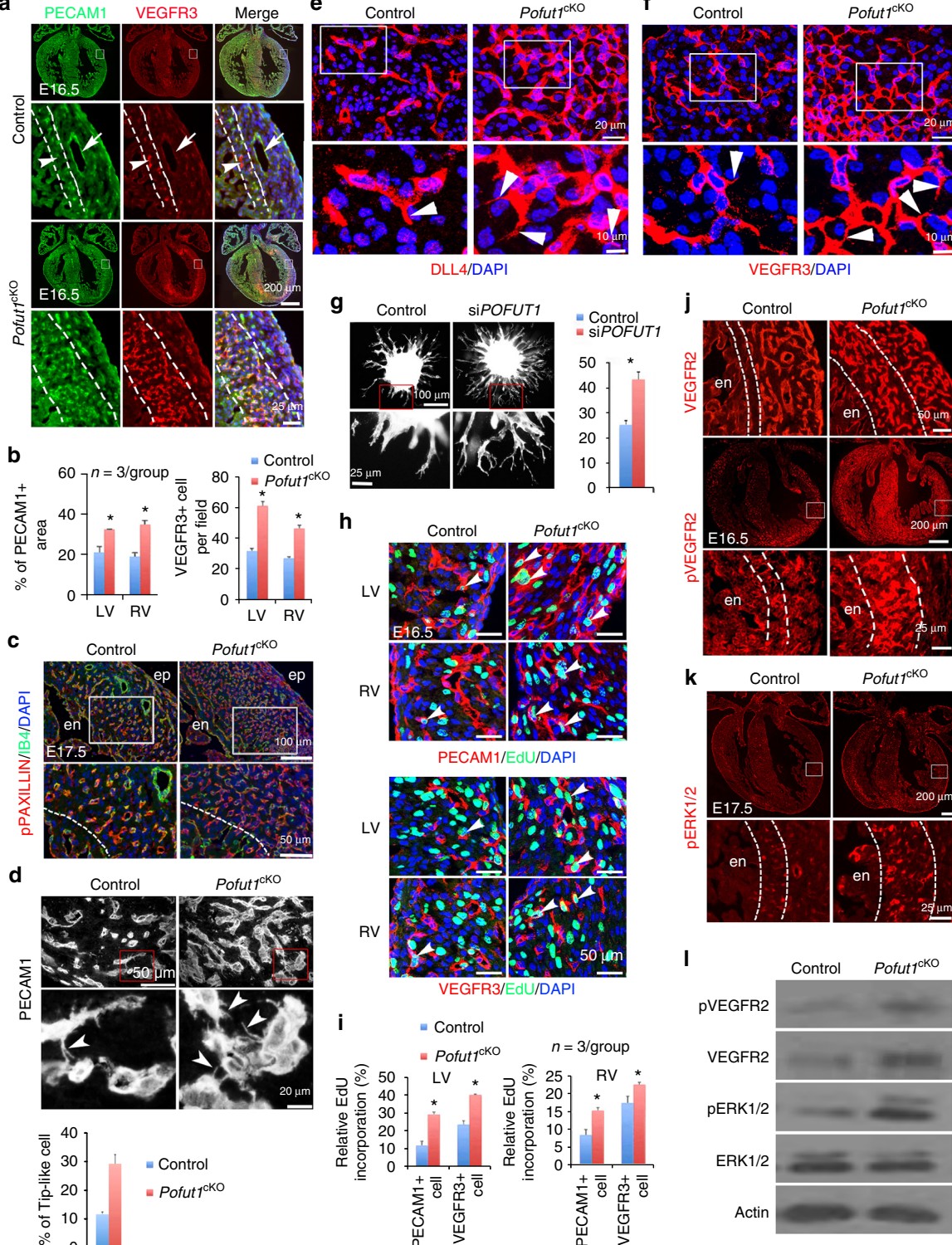

**Fig. 3** POFUT1 regulates angiogenic functions and proliferation of coronary angiogenic cells. **a, b** PECAM1/VEGFR3 co-staining of E16.5 heart sections shows VEGFR3$^{high}$ coronary angiogenic cells (*arrowhead*) in the inner myocardium (*between dotted lines*) and VEGFR3$^{low}$ coronary arteries (*arrow*). **b** Quantification of PECAM1+ areas and the number of VEGFR3$^{high}$ cells. *$p < 0.01$. **c** IB4 and pPAXILLIN co-staining of E17.5 heart sections. en/ep, endocardium/epicardium. **d** PECAM1 immunostaining showing increased tip-like cells (*arrowhead*) in myocardium of E16.5 *Pofut1*$^{cKO}$ embryos. *$p < 0.01$. **e, f** Immunostaining of DLL4 and VEGFR3. **g** Spheroid sprouting assay with phalloidin staining showing increased sprouts by inhibition of *POFUT1* using shRNA. Eight spheroids per group, $n = 3$, *$p < 0.001$. **h, i** EdU labeling for proliferative PECAM1+ or VEGFR3+ cells (*arrowheads*). *$p < 0.05$. All bar charts represent mean ± SD. **j** immunostaining of VEGFR2 and pVEGFR2. **k** Immunostaining of pERK1/2. **l** Western blot analysis of pVEGFR, VEGFR2, pERK1/2, and ERK1/2 expression in E16.5 control and *Pofut1*$^{cKO}$ hearts

performed immunostaining for venous marker EMCN and lymphatic maker LYVE1, respectively. The results showed that some small intramyocardial coronary plexuses in *Pofut1*[cKO] embryos were positive for EMCN while coronary veins on the heart surface were comparable between control and *Pofut1*[cKO] embryos (Supplementary Fig. 6a, b). Whole-mount staining of LYVE1 indicated that anterior cardiac lymphatic vessels were reduced, whereas posterior lymphatics were comparable between control and *Pofut1*[cKO] embryos (Supplementary Fig. 7). These results demonstrate that POFUT1 in the endocardial/coronary endothelial lineage primarily regulates the formation of coronary arteries and may be necessary for maintaining the arterial fate of coronary plexus.

**POFUT1 regulates expansion of coronary angiogenic precursors.** The hypoxia and VEGFA gradient in the inner myocardium suggest that this is the site undergoing vasculogenesis during coronary artery development. Since DLL4 is known to be highly expressed by vascular tip cells[23, 24] during sprouting angiogenesis, an increased number of DLL4-positive coronary plexuses in the inner myocardium of *Pofut1*[cKO] embryos would point to increased vasculogenesis following loss of *Pofut1*. We therefore examined the expression of VEGFR2 and VEGFR3, which are two major angiogenic markers. VEGFR2 was evenly expressed by all coronary endothelium, while VEGFR3 staining revealed two distinct coronary endothelial cell populations in the myocardium of E16.5 hearts (Supplementary Fig. 8a, b). One population expressing high levels of VEGFR3 (VEGFR3[high]) in newly-formed coronary plexuses was located in the inner myocardium. The other population that expressed low levels of VEGFR3 (VEGFR3[low]) was present in the outer myocardium and includes endothelial cells of mature coronary arteries (Fig. 3a, b, Supplementary Fig. 8b). In *Pofut1*[cKO] hearts, the population of VEGFR3[high] cells was greatly expanded (Fig. 3a, b).

We have previously shown that ventricular endocardial cells labeled by *Nfatc1* are a major source of coronary arterial endothelium[7]. To determine whether these VEGFR3[high] cells are derived form endocardium, we performed co-immunostaining of VEGFR3 with *Nfatc1*[Cre] lineage marker GFP from E11.5 to E16.5. VEGFR3 was highly expressed in a subset of endocardial cells at the base of trabeculae, intimately contacting the compact myocardium at E11.5 when the earliest coronary plexus starts to form in the coronary sulcus and interventricular septum (Supplementary Fig. 9). These VEGFR3[high] endocardial cells clearly protruded into the myocardium and represented the earliest endocardial 'sprouts'. The VEGFR3[high] cell population expanded in the myocardium as a coronary plexus after E12.5. At E15.5 (soon after coronary circulation begins around E14.5), the fast growing compact myocardium received active perfusion from newly-established coronary arteries located in the outer layer, which was thus less hypoxic. In contrast, the inner layer distant from the coronary arteries was more hypoxic (Fig. 2e). These findings suggest the inner myocardium is the region continuously undergoing vasculogenesis after the initiation of coronary circulation. Consistently, endothelial cells located in the inner myocardium expressed high levels of VEGFR3, while the endothelial cells in mature arteries located in the outer myocardium, express low levels of VEGFR3 (Fig. 3a, Supplementary Fig. 9). Importantly, lineage tracing and quantification of VEGFR3[high] cells at E16.5 showed that the majority (~ 70%) of VEGFR3[high] cells were derived from *Nfatc1*+ endocardial cells, and this proportion was increased in *Pofut1*[cKO] hearts (Supplementary Fig. 8c, d; Supplementary Fig. 9). To further determine the temporal change of VEGFR3[high] cells in the developing coronary arteries, we compared the VEGFR3 expression in the *Nfact1*+ coronary progeny (GFP labeled) between E13.5 and E16.5. The results showed that >90% GFP-expressing cells expressed high levels of VEGFR3 at E13.5, while this number decreased to 60% at E16.5 (Supplementary Fig. 9). This temporal change was well correlated with VEGFR3[high] angiogenic precursors undergoing vasculogenesis in the forming plexus at E13.5, while reduced VEGFR3 expression was observed in the endothelial cells of mature coronary arteries at E16.5.

We further characterized these VEGFR3[high] cells by co-staining of VEGFR3 with angiogenic markers, including VEGFR2, pVEGFR2 and VEGFA. The results showed that VEGFR3[high] cells were a subpopulation of VEGFR2-expressing cells with high levels of pVEGFR2 and phosphorylated PAXILLIN (pPAXILLIN), which is involved in cell migration by regulating cell focal adhesion turnovers[37, 38] (Supplementary Fig. 10). VEGFA was also highly enriched in the inner myocardium where VEGFR3[high] cells were located (Supplementary Fig. 10). These findings suggest that these VEGFR3[high] endothelial cells were actively migratory and angiogenic. Importantly, like VEGFR3[high] cells, the zone of pPAXILLIN-expressing cells was expanded in *Pofut1*[cKO] hearts (Fig. 3c). At high magnification, PECAM1-, DLL4- or VEGFR3-stained heart sections showed that within the inner myocardium of control hearts, 10% of PECAM1-positive cells had filopodia-like protrusions (microspikes) (Fig. 3d–f)[26, 39], similar to protrusions seem in vascular tip cells. More PECAM1-positive endothelial cells possessed these protrusions in *Pofut1*[cKO] hearts (Fig. 3d). Consistently, in vitro spheroid sprouting assays using HUVECs showed that knockdown of *POFUT1* by shRNA resulted in excessive angiogenic sprouting (Fig. 3g). We next examined the proliferation by EdU incorporation, and found that VEGFR3[high] cells were more proliferative than the overall population of coronary endothelial cells expressing PECAM1, and *Pofut1* deletion promoted proliferation of all endothelial cells (Fig. 3h, i). Together, these results suggest that the VEGFR3[high] cells serve as angiogenic precursor cells for coronary artery formation, and the angiogenic activity of these cells is negatively regulated by POFUT1.

Because hypoxia is known to promote coronary vascular formation through VEGFA/VEGFR2 signaling[7, 40], the inner myocardium is more hypoxic and expresses more VEGFA than the outer myocardium during coronary artery development, and in *Pofut1*[cKO] hearts this hypoxic/VEGFA-expressing zone is greatly expanded (Fig. 2e–h), we examined expression of VEGFR2 and its activated phosphorylated form (pVEGFR2) by immunostaining and western blotting. The result indicated an increased number of VEGFR2-expressing cells and levels of pVEGFR2 in the inner myocardium of *Pofut1*[cKO] hearts (Fig. 3j, l and Supplementary Fig. 11). Furthermore, the phosphorylation of ERK1/2 (pERK1/2), a downstream target of VEGFR2, was elevated in the coronary angiogenic cells in the inner myocardium (Fig. 3k, l and Supplementary Fig. 11). Together, these findings suggest that the increased VEGFR2 signaling may contribute to the over-vasculogenesis phenotype in *Pofut1*[cKO] hearts.

We validated the coronary phenotypes by deletion of *Pofut1* in all vascular endothelium using *Tie1*[Cre] mice. As expected, the coronary phenotype was more severe, leading to embryonic death (Supplementary Table 1). Whole-mount PECAM1 staining indicated absence of the main coronary arteries in hearts of E16.5 pan-endothelial *Pofut1* knockout embryos (Supplementary Fig. 12a). Immunostaining for IB4 and VEGFR3 showed markedly increased coronary plexuses (Supplementary Fig. 12b) and coronary angiogenic cells in the inner myocardium (Supplementary Fig. 12c). Further analysis using SMMHC

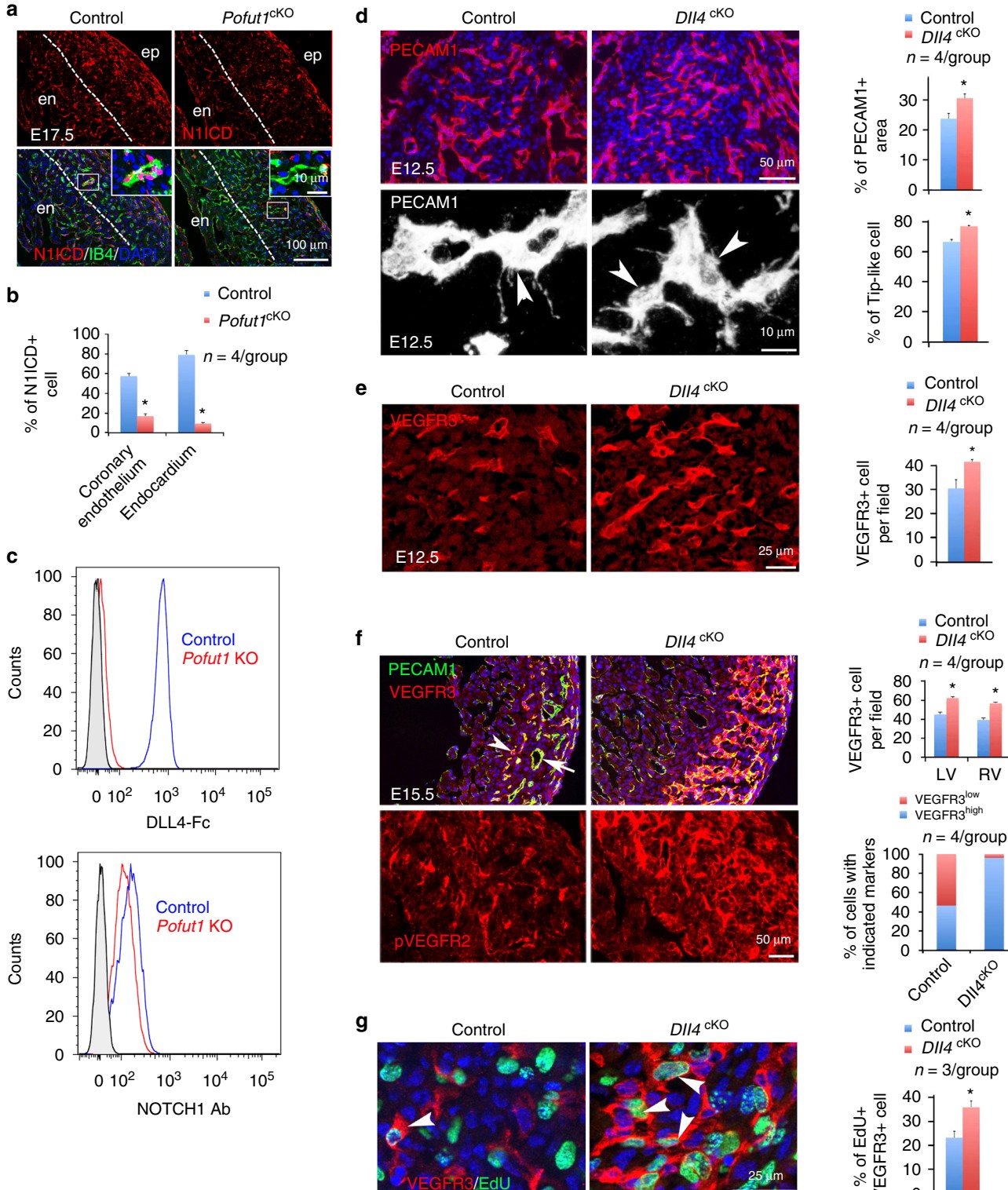

**Fig. 4** POFUT1 regulates coronary artery formation via DLL4/NOTCH1 signaling. **a, b** Co-staining for N1ICD and IB4 showing reduced expression of N1ICD in coronary plexus of E17.5 *Pofut1*cKO heart. *Dotted line* separates trabeculae from compact myocardium. **c** DLL4-Fc binding and NOTCH1 extracellular domain antibody binding to control (*blue* profiles) and *Pofut1* knockout (*Pofut1* KO) (*red* profiles) CHO cells showing reduced DLL4 binding by *Pofut1* KO cells, which have normal NOTCH1 expression. *Grey* profiles reflect secondary antibody alone. **d, e** PECAM1 and VEGFR3 staining for coronary angiogenic cells (*arrowhead*) in coronary plexuses at E12.5, one-day after *Dll4* deletion, showing increased PECAM1+ area, the number of PECAM1+ tip-like or VEGFR3 + cells in *Dll4*cKO (*Dll4*f/f;*Cdh5*CreERT2). **f** PECAM1 and VEGFR3 staining for coronary angiogenic cells (*arrowhead*) and arteries (*arrow*) at E15.5, one-day after *Dll4* deletion, shows increased number of VEGFR3high cells in *Dll4*cKO. *$p < 0.001$. **g** EdU labeling of VEGFR3+ coronary angiogenic cells at E15.5, 1 day after *Dll4* deletion, shows increased proliferation of VEGFR3 cells (*arrowheads*). All bar charts represent mean ± SD; *$p < 0.001$. en/ep, endocardium/epicardium

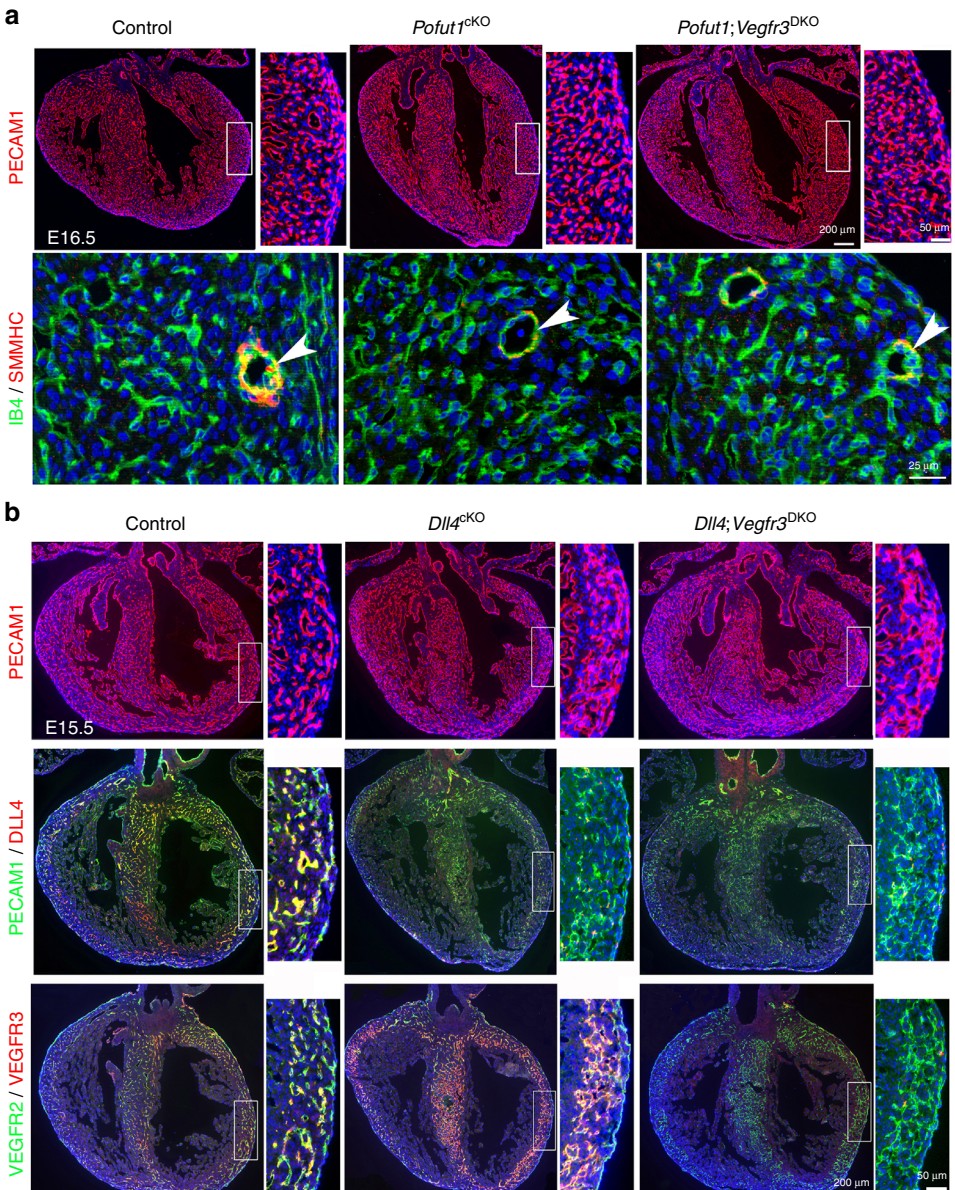

**Fig. 5** Deletion of *Vegfr3* fails to rescue coronary defects in *Pofut1*^cKO or *Dll4*^cKO mice. **a** PECAM1 staining and IB4/SMMHC co-staining shows increased coronaries in E16.5 heart with endocardial *Pofut1* deletion, which cannot be rescued by simultaneous deletion of *Vegfr3*. IB4 and SMMHC co-staining shows defective major coronary arteries (*arrowhead*) remain in the double mutants. **b** PECAM1 staining shows increased coronaries in E15.5 *Dll4*^cKO heart by induced *Dll4* deletion at E14.5, which cannot be rescued by simultaneous deletion of *Vegfr3*. PECAM1/DLL4 or VEGFR2/VEGFR3 co-staining shows effective deletion of *Dll4* or *Vegfr3*

antibodies confirmed no major coronary arteries in these knockout hearts (Supplementary Fig. 12d). Co-immunostaining for PECAM1 with SM22α or αSMA confirmed greatly increased small coronary vessels that were unable to mature into larger arteries (Supplementary Fig. 12e). Consistent with the endocardial deletion of *Pofut1*, the pan-endothelial deletion resulted in significantly increased proliferation of coronary endothelial cells (Supplementary Fig. 12f, g). Altogether, these observations demonstrate that POFUT1 controls coronary artery development by regulating angiogenic functions and proliferation of coronary precursor cells, which express high levels of VEGFR3.

**POFUT1 controls NOTCH signaling in coronary artery formation**. To investigate NOTCH signaling downstream of POFUT1, we examined expression of NOTCH1, N1ICD, and DLL4 in fetal hearts. Endocardial deletion of *Pofut1* did not appear to affect levels of membrane NOTCH1 or DLL4 in individual cells at E14.5 (Supplementary Fig. 13), yet resulted in markedly decreased expression of nuclear N1ICD in endocardium and coronary arterial endothelium at E14.5 and E17.5 (Fig. 4a, b and Supplementary Fig. 13). Ligand binding to CHO cells lacking *POFUT1* had essentially no binding of soluble DLL4-Fc ligand but equivalent expression of NOTCH1 at the cell surface, indicated by binding of antibodies against NOTCH1 extracellular domain (Fig. 4c and Supplementary Table 2). We therefore deleted *Dll4* in endocardial lineage by crossing floxed *Dll4* mice with *Nfatc1*^Cre mice to determine if loss of *Dll4* would mimic loss of *Pofut1* in coronary artery development. However, the deletion was embryonic lethal before E11.5 and the null embryos were already runted around E10.5 (Supplementary

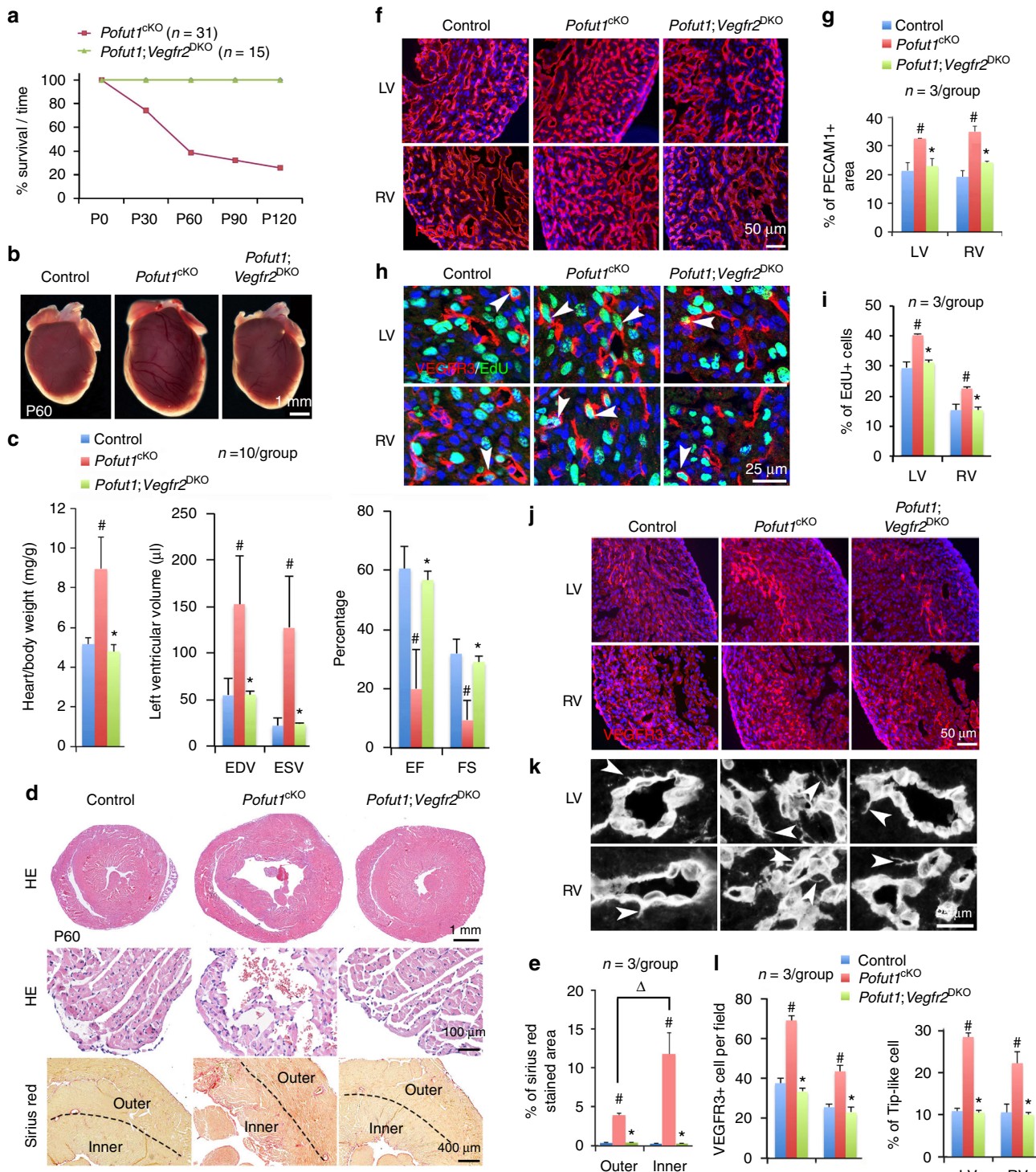

**Fig. 6** VEGFR2 mediates POFUT1 function for coronary artery formation. **a** Survival curve of *Pofut1*cKO and *Pofut1;Vegfr2*DKO mice. **b** Gross view of P60 hearts. **c** Heart/body ratio of P60 mice. #*p* < 0.01 between control and *Pofut1*cKO, *\*p* < 0.01 between *Pofut1*cKO and *Pofut1;Vegfr2*DKO. Echocardiography shows that cardiac function of 2-month *Pofut1;Vegfr2*DKO mice are restored. #,*\*p* < 0.001. EDV/ESV, end diastolic/end systolic volumes; EF/FS, ejection fraction/fractional shortening. **d** Cardiac histology shows no sign of myocardial infarction in P60 *Pofut1;Vegfr2*DKO hearts. **e** Quantification of Sirius Red staining. #*p* < 0.01 between control and *Pofut1*cKO, *\*p* < 0.01 between *Pofut1*cKO and *Pofut1;Vegfr2*DKO. Δ *p* < 0.01 between outer and inner.
**f**, **g** PECAM1 staining shows restored coronary network in E16.5 *Pofut1;Vegfr2*DKO hearts. **h**, **i** VEGFR3/EdU co-staining shows reduced proliferation of angiogenic cells (*arrowheads*) to normal levels in P60 *Pofut1;Vegfr2*DKO hearts. #,*\*p* < 0.05. **j**, **l** VEGFR3 staining shows a restored number and distribution of angiogenic cells in E16.5 *Pofut1;Vegfr2*DKO hearts. Mean ± SD; #,*\*p* < 0.001. **k**, **l** PECAM1 staining shows a restored number of tip-like cells (*arrowheads*) in E16.5 *Pofut1;Vegfr2*DKO hearts. Mean ± SD; #,*\*p* < 0.001

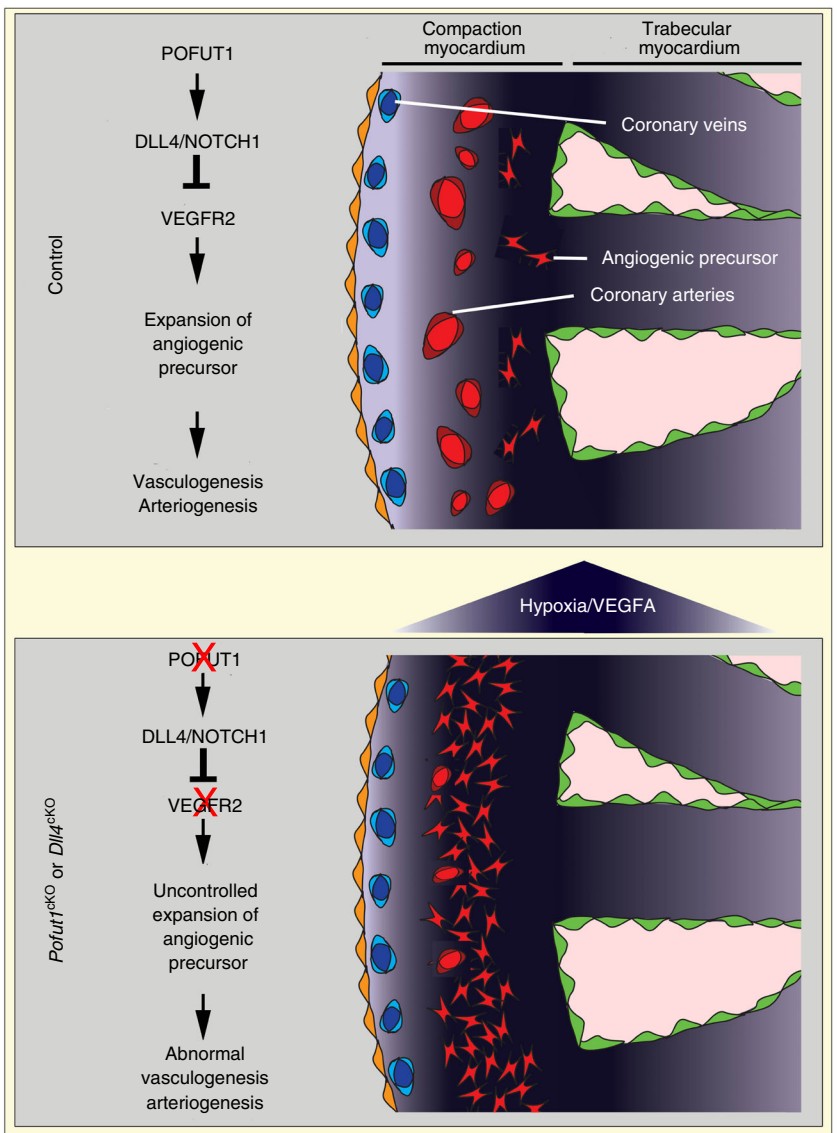

**Fig. 7** POFUT1 regulates proliferation and angiogenic functions of precursor cells for coronary artery development. During coronary artery development, coronary angiogenic precursor cells give rise to coronary arterial plexuses in the hypoxic and VEGFA-rich inner myocardium by vasculogenesis. These plexuses then invade and fuse to form coronary arteries in the outer myocardium through arteriogenesis. Proliferation of coronary angiogenic cells during coronary vasculogenesis is modulated by POFUT1/DLL4/NOTCH1/VEGFA/VEGFR2 signaling axis. Deletion of *Pofut1* suppresses DLL4/NOTCH1 signaling, which in turn elevates VEGFA/VEGFR2 signaling. Augmented VEGFA/VEGFR2 signaling promotes proliferation and angiogenic functions of coronary precursor cells, resulting in excessive non-functional coronary plexuses, which are unable to mature into coronary arteries

Fig. 14). To overcome the early lethality, we used the inducible endothelial Cre line Cdh5(PAC)/CreERT2[41] to delete *Dll4* at E11.5 or E14.5. PECAM1 and VEGFR3 co-immunostaining showed that deletion at either time point resulted in increased formation of coronary angiogenic cells and coronary plexuses (Fig. 4d–f). Of note, the deletion of *Dll4* at E14.5 appeared to promote VEGFR3[Low] endothelial cells in the outer myocardial wall to become VEGFR3[high] angiogenic cells (Fig. 4f, Supplementary Fig. 15). In addition, loss of *Dll4* decreased NOTCH signaling and augmented pVEGFR2 and the proliferation of VEGFR3[high] cells (Fig. 4f, g, Supplementary Fig. 16), similar to lack of DLL4 binding to NOTCH1 due to deletion of *Pofut1*.

NOTCH inhibition by the gamma-secretase inhibitor DAPT may increase microtubule dynamics[42], and increased microtubule dynamics is correlated with increased cell migration[43]. We therefore sought to determine whether inhibition of NOTCH signaling would promote VEGFA-induced cell migration in cultured HUVEC cells. The results of time-lapse imaging showed that either DAPT treatment or shRNA knockdown of *POFUT1* or *DLL4* significantly increased cell mobility indicated by increased velocity and mean square displacement (Supplementary Fig. 17a–d). Co-staining for acetylated and total microtubules showed that inhibition of NOTCH signaling decreased the ratio of acetylated to total microtubules (Supplementary Fig. 17e, f), indicating increased microtubule plasticity as acetylation of microtubules is associated with more stable, long lived and less dynamic microtubules[44]. The increased cell mobility was correlated with an increased number of filopodia (Supplementary Fig. 17g, h). These findings support the regulation of coronary angiogenic cell migratory behavior by POFUT1 through regulating DLL4/NOTCH1-induced signaling in coronary artery formation.

**VEGFR2 mediates POFUT1 function in coronary artery formation**. Since *Pofut1* or *Dll4* deletion promoted VEGFR3 expression and increased VEGFR2 phosphorylation, we next investigated whether signal alterations through VEGFR3 and VEGFR2 are responsible for the coronary phenotype in *Pofut1*<sup>cKO</sup> or *Dll4*<sup>cKO</sup> embryos. We first investigated the role of *Vegfr3* and *Vegfr2* in coronary artery development by deleting either gene using the *Nfatc1*<sup>Cre</sup> driver. The results showed that VEGFR3 suppresses and VEGFR2 promotes coronary vasculogenesis (Supplementary Fig. 18a–c). To directly test whether VEGFR3 and VEGFR2 work downstream of NOTCH signaling in coronary artery development, we simultaneously deleted *Vegfr3* and *Dll4*, or *Vegfr3* and *Pofut1*. Consistently, inactivation of *Vegfr3* did not alleviate the coronary phenotype resulting from *Dll4* or *Pofut1* deletion (Fig. 5a, b, Supplementary Fig. 18d, e). Strikingly, however, inactivation of VEGFR2 fully rescued the lethality of *Pofut1*<sup>cKO</sup> mice (Fig. 6a). Compared to their littermates with deletion of *Pofut1*, mice with double deletion of *Pofut1* and *Vegfr2* (*Pofut1;Vegfr2*<sup>DKO</sup>) had normal heart size (Fig. 6b, c) and cardiac function, including end-diastolic and end-systolic volumes, as well as ejection fraction and fraction shortening (Fig. 6c). Histologically, *Pofut1; Vegfr2*<sup>DKO</sup> mice had no sign of myocardial infarction by HE staining, nor was there increased cardiac fibrosis by Sirius Red staining (Fig. 6d, e). Immunostaining for PECAM1 revealed that *Pofut1;Vegfr2*<sup>DKO</sup> embryos had coronary vasculature that was comparable to controls (Fig. 6f, g). Co-immunostaining for VEGFR3 and EdU indicated that the coronary vasculature in *Pofut1;Vegfr2*<sup>DKO</sup> embryos contained angiogenic cells with a normal proliferative rate (Fig. 6h, i). Consistently, *Pofut1; Vegfr2*<sup>DKO</sup> embryos had a normal number of VEGFR3<sup>high</sup> angiogenic precursor cells (Fig. 6j–l). These observations indicate that augmented VEGFR2 signaling in coronary angiogenic precursors of *Pofut1*<sup>cKO</sup> mice likely drives the overall coronary phenotype, including increased proliferation of coronary angiogenic precursors and formation of coronary plexuses, with defective arteriogenesis.

## Discussion

Collectively, our studies suggest that VEGFR3<sup>high</sup> coronary angiogenic precursors undergo vasculogenesis to form coronary plexuses, which subsequently mature into large coronary arteries by arteriogenesis (Fig. 7). In this sequential process, POFUT1 regulates the proliferation of coronary angiogenic precursors and their functions through regulating the DLL4/NOTCH1/VEGFA/VEGFR2 signaling axis. Disruption of *Pofut1*, which controls the strength of NOTCH ligand binding, or *Dll4*, results in coronary artery anomalies that lead to early-onset ischemic heart disease. We identify the primary early developmental defect as increased angiogenic cell proliferation and function, which promotes coronary vasculogenesis, but impairs subsequent arteriogenesis. The latter maturation defect is supported by defective recruitment of smooth muscle cells during arteriogenesis[28–30, 45]. The defective coronary arteries fail to provide sufficient perfusion to meet the demands of the growing heart and cardiac functional output, resulting in severe myocardial hypoxia, and inevitably myocardial infarction and heart failure. In some cases, rupture of large coronary aneurysms causes large myocardial infarctions and early death of young mice.

POFUT1 modifies NOTCH receptors by transferring O-fucose to their epidermal growth factor (EGF)-like repeats[46]. This post-translational modification is essential for proper NOTCH ligand–receptor interactions and subsequent signaling[33–35, 47]. Notch signaling is well known to inhibit sprouting angiogenesis during retinal and tumor angiogenesis[23, 24, 48, 49]. FRINGE, an enzyme that also modifies NOTCH receptors after POFUT1 modification, leads to preferential DLL4 binding to repress retinal angiogenesis[25]. However, whether POFTU1 is required for angiogenesis or vascular development has not yet been determined. This study for the first time documents that POFUT1 is required for coronary artery formation through regulating vasculogenesis and arteriogenesis. Like FRINGE, POFUT1 promotes NOTCH signaling through DLL4 binding[25].

We have previously shown that endocardium labeled by *Nfatc1* has a major contribution to coronary arterial endothelium[7]. Deletion of *Pofut1* in *Nfatc1*<sup>+</sup> lineage severely impairs coronary artery development and has minor effects on coronary vein or lymphatic vessel development. To better explain the cell type-specific phenotype caused by *Pofut1* deletion, the contribution of the *Nfatc1*<sup>+</sup> lineage to different coronary vessels was analyzed by co-immunostaining of cell type-specific markers with the *Nfatc1*<sup>+</sup> lineage reporter, and the experiments were carried out independently in Zhou and Harvey laboratories. The results confirm that the *Nfatc1*<sup>+</sup> endocardial lineage has a major contribution to embryonic coronary arteries (>70%), a minor contribution to coronary vein (<50%), and no, or very little, contribution to cardiac lymphatic endothelium (Supplementary Figs. 19–21).

Our study has identified two distinct endothelial cell subpopulations in developing coronary arteries. They are VEGFR3<sup>low</sup> cells in the mature coronary arteries in the outer myocardium, and VEGFR3<sup>high</sup> cells in the primitive coronary plexus within the hypoxic inner myocardium where VEGFA is enriched. These VEGFR3<sup>high</sup> cells also express high levels of DLL4. Both VEGFR3 and DLL4 are highly expressed in vascular tip cells of developing vessels in the retina and angiogenic tumor vessels[50]. Furthermore, VEGFR3<sup>high</sup> cells are a subpopulation of VEGFR2-expressing cells with high levels of pVEGFR2, suggesting that they are actively angiogenic. PAXILLIN is a downstream target of VEGFA signaling and regulates endothelial cell migration by modulating focal adhesion turnover[37, 38]. Our results show that VEGFR3<sup>high</sup> and VEGFR3<sup>low</sup> cells have high and low levels of pPAXILLIN, respectively, suggesting that VEGFR3<sup>high</sup> cells are more mobile. In addition, co-immunostaining of VEGFR3 with the *Nfatc1*<sup>Cre</sup>-mediated GFP lineage reporter reveals that VEGFR3<sup>high</sup> cells are derived from the endocardium. Together, our findings suggest that the VEGFR3<sup>high</sup> cells arise from the endocardium and serve as an angiogenic precursor cell population for coronary artery development. Further, their angiogenic feature is controlled by POFUT1 via NOTCH signaling.

Angiogenic precursor cells form vascular plexuses through vasculogenesis and this process is regulated by NOTCH and VEGFA signaling[23, 25, 51]. Our study indicates that during formation of embryonic coronary arteries, VEGFR3<sup>high</sup> cells may represent a progenitor cell pool for continuous coronary artery formation to meet the increasing demands of the growing myocardium. Our model predicts that in normal coronary artery formation, VEGFR3<sup>high</sup> cells form plexuses in the inner myocardium and then organize into mature vessels under the regulation of a hypoxia-driven VEGFA gradient via VEGFR2. DLL4/NOTCH1 interactions among the VEGFR3<sup>high</sup> cells modulate their proliferation, migration and maturation by suppressing VEGFA/VEGFR2 signaling. With depressed DLL4/NOTCH1 interactions caused by *Pofut1* inactivation, VEGFA/VEGFR2 signaling intensifies in the VEGFR3<sup>high</sup> angiogenic cells, leading to expansion of this progenitor pool by increased proliferation, as well as an increase in angiogenic migration, possibly driven by an increased hypoxic stimulus. However, the excessive coronary angiogenic cells only form immature, non-functional plexuses. This promotes further increased hypoxia in the ventricular myocardium leading to

increased VEGFA activity that amplifies the already dysregulated vasculogenesis by inducing the formation of more immature VEGFR3[high] angiogenic cells.

Deletion of *Pofut1* or *Dll4* also results in defective coronary arteriogenesis. JAG1/NOTCH3 signaling is known to promote smooth muscle cell recruitment and differentiation during vascular maturation[28–30, 45] and both genes are significantly down-regulated by deletion of *Pofut1*. However, our rescue experiment would favor the notion that defective arteriogenesis is secondary to the early vasculogenesis defect, since disruption of *Vegfr2* specifically in endocardium fully rescues both vasculogenesis and arteriogenesis defects in *Pofut1*[cKO] mice. Additionally, deletion of either *Pofut1* or *Dll4* results in expansion of coronary angiogenic cells, which are unable to become mature endothelial cells. Therefore, we suggest that disruption of the DLL4/NOTCH1/ VEGFA/VEGFR2 signaling axis by deletion of *Vegfr2* rescues arteriogenesis defects in *Pofut1*[cKO] mice by repressing the number of angiogenic cells formed and allowing their conversion to mature vessels.

A potential redundant function between VEGFR2 and VEGFR3 in driving angiogenesis downstream of NOTCH signaling has been noted previously. VEGFR3 stimulation augments VEGFA-induced angiogenesis in the presence of VEGFR2 inhibitors, whereas antibodies against both VEGFR2 and VEGFR3 additively inhibit angiogenesis[50]. Similarly, VEGFR3 up-regulation resulting from NOTCH signaling inhibition is able to drive retina angiogenesis without VEGF-VEGFR2 signaling[52]. Clinically, the inhibitors of VEGFA/VEGFR2 signaling show initial efficacy in treatment of tumors, but without lasting effects, which is thought to be due to a compensation from VEGFA/ VEGFR3 signaling[53]. However, our data do not support such compensation in the developing coronary arteries. Rather, they indicate that DLL4/NOTCH1/VEGFA/VEGFR2 is an essential molecular signaling axis underlying coronary artery development. Further investigation is needed to identify the non-redundant roles of VEGFR2 and VEGFR3 in the process, since they may function in a context-dependent manner[46, 50, 52–54].

The new mouse model of coronary artery anomalies and early-onset ischemic heart disease we describe herein will provide novel opportunities for gaining deeper insights into the developmental origins and genetic mechanisms of coronary vessel development. The information may have clinical implications in developing genetic screening methods for coronary artery anomalies and providing a developmental paradigm for regenerative medicine for coronary artery disease.

## Methods

**Mouse strains**. Mouse housing and experiments were performed according to the protocols approved by the Institutional Animal Care and Use Committee of Albert Einstein College of Medicine. The tissue specific *Pofut1* knockout mice were generated by crossing floxed *Pofut1* mice[35] with *Tnnt2*[Cre] for cardiomyocytes[55], *Tbx18*[Cre] for epicardium[56], *Sm22a*[Cre] for smooth muscle cells[57], *Mef2c*[Cre] for secondary heart field[58], *Tie1*[Cre] for pan-endothelial cells[59] and *Nfatc1*[Cre] for endocardial lineage[7]. Similarly, endocardial lineage *Dll4* knockout mice were generated by crossing floxed *Dll4* mice[60] with *Nfatc1*[Cre] mice. For inducible deletion of *Dll4* in endothelial cells, inducible Cre line Cdh5(PAC)-CreERT2[41] was crossed with floxed *Dll4* mice. To induce the deletion, tamoxifen was injected into pregnant female mice through IP at a specified time point, and the coronary arteries were analyzed at indicated time points. Endocardial lineage *Vegfr2* or *Vegfr3* knockout mice were generated by crossing floxed *Vegfr2* or *Vegfr3*[61] mice with *Nfatc1*[Cre] mice, respectively. *Pofut1* and *Vegfr2* endocardial lineage double knockout mice were generated by crossing floxed *Pofut1* and floxed *Vegfr2* mice with *Nfatc1*[Cre] mice. *Pofut1* and *Vegfr3* endocardial lineage double knockout mice were generated by crossing floxed *Pofut1* and floxed *Vegfr3* mice with *Nfatc1*[Cre] mice. Inducible *Dll4* and *Vegfr3* endothelial double-knockout mice were generated by crossing floxed *Dll4* and floxed *Vegfr3* mice with Cdh5(PAC)-CreERT2 mice. Floxed *Pofut1* mice were maintained in mixed background, while all other mouse stains were in C57B6 background. Embryos were isolated and inspected according to expected developmental ages. Underdeveloped embryos were excluded from studies. Age matched adult mice and embryos were randomly selected during

experiments. Noontime on the day of detecting vaginal plugs was designated as E0.5. The yolk sac or tail was used for PCR genotyping.

**Measurement of cardiac function by echocardiography**. Echocardiography was performed at the Einstein Cardiovascular Physiology and Imaging Core Facility to evaluate the cardiac functions of control, *Pofut1*[cKO], and *Pofut1;Vegfr2*[DKO] mice of both sexes at two-month old. Investigators were blinded to genotypes of mice and subsequent data analysis. Mice were anesthetized by inhalation of 2% isoflurane in glass chamber and 1 to 1.5% isoflurane via a nose cone. Echocardiographic assessment was performed using a Vevo 770 ultrasound with a real time micro-visualization transducer applied parasternally to the shaved chest wall. Vevo 770 quantification software was used to measure left ventricular end-diastolic volume (LVEDV), left ventricular end-systolic volume (LVESV), the ejection fraction (EF) and fractional shortening (FS). Statistical significant comparisons among groups were performed by one-way analysis of variance following Turkey's post-test. $p < 0.05$ was considered significance.

**Micro-computed tomography (MicroCT)**. MicroCT was performed to access the coronary artery in P20 control and *Pofut1*[cKO] mice as described previously[62]. Investigators were blinded to genotypes of mice and subsequent data analysis. In brief, the freshly euthanized mice were injected with bismuth contrast solution into descending aorta before the heart stops beating. Mice were immediately chilled in ice and immersion fixed in 2% paraformaldehyde (PFA) overnight. 2D microCT scans were acquired with a GE eXplore Micro-CT System (GE Healthcare) using a 400 cone beam filtered back projection algorithm, set to an 8–27-µm micron slice thickness. Data were acquired in an axial mode, covering a volume of 2.0 cm in the z direction with a 1.04-cm field of view. During post-processing, a 40,000 gray-scale value was set as a threshold to eliminate noise with minimal sacrifice of vessel visualization. The micro-CT data were processed using real-time 3D volume rendering software and Microview software to reconstruct three 2D maximum-intensity projection images ($x$, $y$, $z$ axes) from the raw data. Quantification was performed using a modified Image ProPlus 5.0 algorithm (Media Cybernatics). Data are expressed as vessel number, representing total number of vessels, of specified diameter counted in 350–400 z sections.

**Histology and immunostaining**. The preparation of paraffin and frozen sections and immunostaining were carried out following standard protocol as described previously[63]. Briefly, for paraffin sections, embryonic or adult hearts were freshly isolated in PBS, fixed in 4% PFA at 4 °C overnight, washed in PBS, dehydrated through a serial of gradient ethanol, cleared in xylene and embedded in paraffin. The hearts were then cut at 6 µm and the tissue sections were mounted on positive charged slides. HE and Sirius red staining were performed using the standard protocol to evaluate the histology and pathology of the heart. For frozen sections, the hearts were isolated freshly, fixed in 4% PFA at 4 °C for a few hours dependent on the size of the tissues, soak in 15% and 30% sucrose sequentially and embedded in OCT compounds with orientation for front sections. The hearts were then cut at 8 µm, mounted on positive charged slides, post-fixed in cold ethanol and acetone (1:1) solution for 5 min and stored at −80 °C. Tissue sections were air dried for 45 min before staining. Tissue sections were blocked with 5% normal horse or goat serum for 1 h at room temperature (RT) and incubated with primary antibodies with indicated dilution in blocking buffer at 4 °C overnight. Then the tissues were incubated with secondary antibodies conjugated with the Alexa 568 or 488 fluorescence dyes for 1 h at RT. To visualize the tip-like cell morphology (the microspikes or filopodia-like protrusion structure), the tyramide signal amplification system (PerkinElmer) was used for PECAM1 antibody staining. Antibody information was listed in Supplementary Table 3. Images were taken by a Zeiss Observer Z1 or Leica SP5 confocal microscope. All imaging quantifications were carried out blindly using Image J.

**Detection of myocardial hypoxia**. Hypoxia in embryonic or postnatal hearts was detected using Hypoxyprobe Kit (Hypoxyprobe Inc). The pregnant female mice or postnatal mice were injected with Hypoxyprobe[TM]-1 through intraperitoneal (IP) at a concentration of 60 mg/kg body weight After 90 min pulse, the hearts were isolated and processed for frozen section as described above. Then the hypoxia was detected by immunostaining of Hypoxyprobe antibody and Alexa Fluor 488 goat anti-mouse IgG secondary antibody. The stained tissues were photographed using a Zeiss Observer Z1 microscope.

**Whole-mount staining and coronary arteriography**. To visualize the coronary vessels in embryonic hearts, whole-mount staining was performed using antibodies against PECAM1, EMCN or LYVE1 as described previously[64]. Briefly, the hearts were isolated from E16.5 control and *Pofut1*[cKO] embryos, and fixed in 4% PFA for 2–3 h at 4 °C. The hearts were then blocked in 5% normal horse serum for 3 h at 4 °C and followed by incubation with primary antibodies at 4 °C overnight. The hearts were then washed for five times with PBS containing 0.5% TritonX-100 and incubated with biotinylated secondary antibodies at 4 °C overnight. The hearts were then washed for five times with PBS containing 0.5% TritonX-100 and incubated with streptavidin from ABC Kit (Vector Laboratories) at 4 °C overnight. The hearts were then washed for five times with PBS containing 0.5% TritonX-100

and developed color using DAB Kit (Vector Laboratories). In addition, coronary angiogram was performed to access the embryonic coronary arteries by injecting fluorescence dye into the left ventricle of beating hearts. The dye entered and labeled the coronary arteries through the coronary ostia (the coronary openings at the root of aorta). After injection the hearts were collected, fixed in 4% PFA, and photographed using a Zeiss Discovery Microscope.

**Quantification of vessel density and VEGFR3$^{high}$ coronary angiogenic cells**. All quantifications were carried out using Image J. For quantifying the vessel area, images from PECAM1 antibody staining were used. The images from VEGFR3 antibody staining were used to quantify the number of VEGFR3+ cells. To quantify tip-like angiogenic cells, the heart sections were stained with PECAM1 and PECAM1-positive cells with microspikes were counted. The data were presented as the ratio of tip cells among total endothelial cells. Four hearts were analyzed for each experiment.

**Spheroid angiogenic sprouting assay**. Spheroid sprouting assay was performed as described previously[65]. Briefly, HUVEC (Life Technology) was cultured in M200 media. The cells were infected with Lentivirus carrying non-silence shRNA or POFUT1 shRNA and the infected cells were enriched by adding 2 µg/ml of puromycin to the media. Four thousand cells were cultured in round-bottom 96-well plates, which pre-coated with 0.8% agarose for spheroid formation. The spheroids were then embedded in fibrin gel and cultured in M200 media containing 10 ng/ml of VEGF for 48 h. Then the spheroids were fixed in 4% PFA for 1 h and stained with phalloidin and imaged using EVOS microscope. The sprouting number was counted manually using Image J software. Eight spheroids were analyzed for each group in each experiment. The data from three independent experiments were subjected to statistic calculation.

**Cell proliferation and apoptosis assays**. Cell proliferation and cell apoptosis were determined using EdU and TUNEL assays, respectively, as described previously[63]. For cell proliferation assay, pregnant female mice were injected with EdU (Life Technology) through IP at a concentration of 100 mg/kg. After a 2-hour pulse, the hearts were collected and processed for frozen sections as described above in Immunostaining Section. The serial sections crossing the whole heart were first stained with PECAM1 or VEGFR3 antibody followed by EdU staining with EdU imaging Kit (Life Technology) and counterstained with DAPI (Vector lab). The stained sections were photographed using a Zeiss Observer Z1 or Leica SP5 confocal microscope. EdU and PECAM1 or VEGFR3 double positive cells were counted using Image J and the data was presented as the ratio of EdU-positive cells among total PECAM1- or VEGFR3-positive cells. Three hearts were analyzed for each genotype. Apoptotic cells in the developing hearts of E14.5 embryos were visualized by TUNEL assay. The frozen sections of isolated hearts were prepared as described in the EdU study. Serial sections were first stained with PECAM1 antibody, followed by TUNEL assay by using DeadEnd™ Fluorometric TUNEL System (Promega) and counterstained with DAPI. The stained sections were photographed using a Zeiss Observer Z1 or Leica SP5 confocal microscope.

**RNA extraction and quantitative PCR (qPCR)**. To assess gene expression in mouse embryonic hearts, total RNA was extracted from embryonic hearts using the TRIzol solution (Life Technology) for reverse transcription to generate cDNAs with a Superscript II reverse transcriptase Kit (Life Technology). qPCR was performed using the Power SYBR Green PCR Master Mix (Life Technology) containing gene-specific primers (Supplementary Table 4). The relative expression of each gene was normalized to the expression of *Gapdh* and calculated using the 2$^{-\Delta\Delta CT}$ method. Biological replicates were performed using three individual samples of each genotype and technical triplicates were carried out for each run of qPCR. Student's *t*-test was used for statistic comparison between groups. *p* < 0.05 was considered as significant.

**Generation of *Pofut1* knockout cells using CRISPR/Cas9**. Lec1 CHO cells were transduced with Lentivirus carrying doxycycline-inducible Cas9 and selected with 20 µg/ml of puromycin. After 3 days, the cells were transduced with a second Lentivirus containing the guide RNA (gRNA) (5′-GGCGGCGCCCATGTCGGC GG-3′), which targets the second coding exon of *Pofut1*. After selection for 7 days in 25 µg/ml of blasticidin, resistant cells were maintained in the presence of both puromycin and blasticidin, and Cas9 expression was induced by addition of 500 ng/ml doxycycline. After 7 days individual cell clones obtained by limiting dilution were analyzed by T7 endonuclease assay and DNA sequencing. Two independent clones having a deletion of two or four nucleotides at the Cas9-gRNA site were shown to lack POFUT1 by western analysis with rabbit anti-bovine POFUT1 antibody[16]. Cells not treated with doxycycline were used as controls.

**DLL4-Fc binding assay and NOTCH1 cell surface expression**. Binding assays were performed as described previously[66]. Briefly, cells were washed with ligand-binding buffer (Hanks' buffered salt solution, pH 7.4, 1 mM CaCl₂, 1% (w/v) bovine serum albumin, 0.05% NaN₃), fixed in 4% PFA in phosphate buffered saline without divalent cations (PBS-CMF) at RT for 10 min and stored in ligand binding buffer at 4 °C. Fixed cells (1 × 10⁶) were incubated with 100 µl ligand binding buffer containing 7.5 µg/ml DLL4-Fc or anti-hamster NOTCH1 antibody (R&D Systems, AF5267, 1:50 in ligand binding buffer) at 4 °C for 1 h. Soluble DLL4-Fc ligand was prepared as described previously[66]. After incubation, cells were washed twice with 0.5 ml ligand binding buffer and incubated with phycoerythrin (PE)-conjugated goat anti-human IgG (Fc-specific) or Rhodamine Red-X-conjugated donkey anti-sheep IgG (1:100 in ligand binding buffer) at 4 °C for 30 min in the dark. The cells were then washed twice with 1 ml of ligand binding buffer and analyzed in a FACS Calibur flow cytometer.

**Cell migration assays**. Cell migration studies were performed using HUVECs (Life technologies). 3 × 10⁴ cells were plated onto 24-well plates. For inhibition of NOTCH signaling, 20 µM DAPT was added to the media, while DMSO was used as controls. For knockdown of *POFUT1* or *Dll4*, the cells were infected with lentivirus carrying shRNA targeting *POFUT1* or *Dll4* respectively. The infected cells were enriched by puromycin selection. Knockdown efficiency was determined by qPCR. In cell migration assays, 10 ng/ml of VEGF120 was added to the media. The cells were subjected to live imaging using EVOS FL Auto Cell Imaging System. Analysis of individual cell tracks was performed using the DiPer Excel Macro[67]. For immunofluorescence, cells were plated onto glass coverslips. After 24-hour treatment, cells were fixed for 10 min with fixative solution (4% paraformaldehyde, 0.1% Triton X, and 0.15% glutaraldehyde in BRB80) and stained with rabbit anti-tubulin (Abcam, Ab18251), mouse anti-acetylated tubulin (Sigma, T6793) or actin (Alexa Fluor Phalloidin 488, A12379) antibodies for 1 h at RT. Then the cells were incubated with secondary antibodies conjugated with the Alexa 568 or 488 fluorescence dyes for 1 h at RT. Images were acquired using an EVOS FL Auto Cell Imaging System.

**Statistical analysis**. No statistical methods were used to predetermine sample size. Student's *t*-test (two-tailed) or one-way analysis of variance with Turkey's post-test was used for statistical difference between groups, assuming unequal variance from at least three independent experiments. Normality was assumed and variance was compared between groups. Sample size was determined based upon previous experience in the assessment of experimental variability. The investigators were not blinded to the group allocation during experiments and outcome assessment, unless stated otherwise. We chose the adequate statistic tests according to the data distribution to fulfill test assumptions. All numerical data were presented as mean ± SD and *p*-value of <0.05 was considered as significant.

**Data availability**. The data supporting the findings of this study are available within the article and its Supplementary Information files and from the corresponding author on reasonable request.

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

## Acknowledgements

We thank W. Lui for assistance with embryonic coronary arteriography. This work was supported by the National Institutes of Health/National Heart, Lung, and Blood Institute R01HL116997 and R01HL133120 (B.Z.), and the National Institute for General Medical Sciences R01GM106417 and the Albert Einstein Cancer Center grant NCI P0113330 (P. S.).

## Author contributions

Y.W., B.W., P.L, D.Z., and B.W.: Performed most experiments. Z.Z.: Performed microCT. S.V.: Performed binding assays. R.C. and A.K. performed live imaging and quantification of microtubule expression. D.S., G.dMN., N.S., N.F., R.H., R.K., and P.S.: Helped data analysis. R.A. and K.A.: Provided the Cdh5CreERT2 and floxed Vegfr3 mouse line, respectively. Y.W. and B.Z.: Conceptualized the project. Y.W., B.W. and B.Z.: Supervised experiments. Y.W. and B.Z.: Wrote the manuscript. G.dMN., N.S., N.F., R.K., D.S., R.H., and P.S.: Critically edited the manuscript. All authors reviewed and approved the manuscript.

## Additional information

**Competing interests:** The authors declare no competing financial interests.

