## [Peer Review File · Nature Communications]

Reviewers' Comments:

Reviewer #1 (Remarks to the Author):

In this study Wang and colleagues target POFUT1, a modifier of Notch signalling, in the coronary endothelium to reveal a role in angiogenesis and arteriogenesis and identify a putative VEGFR3⁺ angioblast population, which they suggest may act as a progenitor pool for continuous coronary artery formation.

Notch signalling has previously been implicated in coronary artery development, for example Notch 1 has been shown to be important for coronary artery EC commitment and vessel wall maturation (del Monte et al., 2011) and Akt/mTOR acting on Jagged 1 is required for VSMC maintenance such that hearts with loss of Akt and downstream Notch signalling fail to form coronary arteries and arterioles (Kerr et al., 2016). The novelty of the current study, therefore, lies in the targeting of an alternate component of the Notch pathway, POFUT1, and in ascribing function specific to coronary artery formation via misregulated angiogenesis at the level of a putative VEGFR3⁺ progenitor population residing in the compact myocardium. As such this is an interesting study and one which provides insight into how defective arteriogenesis during development may result in ischaemic heart disease including myocardial infarction and heart failure during adulthood. That said, there are some issues with the study as it currently stands and the data as presented do not fully support the conclusions:

General comments:

The authors have not examined the coronary veins and instead have focused exclusively on arteries. Some of the analyses presented do not discriminate between veins and arteries for e.g., Fig. 2a and the whole mount PECAM staining in Fig. 4a, and indeed there may well be effects on the more superficial veins according to this data. Notch signalling when impaired is known to lead to arterialisation of veins and consequently an effect on the coronary veins should be investigated and if nothing else excluded from the *pofut1*CKO phenotype.

More importantly a major proposed finding in this study is the identity of the so-called VEGFR3^{High} sub-population which is suggested to function as a tissue-specific angiogenic population for coronary artery development. The evidence for this is somewhat weak in its current form, with the primary assays based on immunostaining which may simply reflect an up-regulation of VEGFR3 in angiogenic sprouts (as opposed to evidence of a discrete progenitor pool) and which in turn is used (inappropriately) to quantify levels of VEGFR3. Furthermore, the authors defined two angiogenic coronary EC populations based on VEGFR3 expression levels – VEGFR3^{Low} EC in the outer myocardial wall and VEGFR3^{High} EC in the inner myocardium, but it is not clear from their data if both populations are derived from the endocardium or

whether these populations represent different states of the same cell type, i.e. angioblast-like cell type expressing high VEGFR3 and angioblast-derived arterial EC expressing low VEGFR3. Likewise, information is lacking as to when this population(s) arises in the developing myocardium, since the authors focused their studies at E16.5. The term “angioblast” may be inappropriately assigned here and requires detailed lineage tracing data, both to reveal the origin of this “angioblast” population and to trace progenitors and their derivatives into developing coronary beds undergoing angiogenesis and/or coalescence into arteries/arterioles within the compact myocardium. Given the potential significance of this finding and that it represents a possible major advance beyond previously published studies, this population needs more detailed characterisation. This should include either an endocardial lineage tracing approach (e.g. based on the expression of *Nfatc1* reporter) combined with immunostaining/flow cytometry or analysis of VEGFR3-CreERT2; reporter fate mapping during coronary vascular development (VEGFR3-CreER2 mice were recently reported by Martinez-Corral et al., 2016), which in addition to labelling in situ, would also facilitate isolation of the VEGFR3+ population(s) –High v Low reporter expression and transcriptome analyses to determine artery-specific cell fates etc.

Specific comments:

1. The study lacks a comprehensive analyses of the spatio-temporal expression of *Pofut1* in the developing heart to underpin the targeting strategies proposed and to illustrate expression in coronaries (arteries and/or veins). As it stands, the characterization of *POFUT1* expression is incomplete; is *POFUT1* expressed only in the developing endocardium/endothelium? The data shown in Supplementary Fig. 1a is suboptimal and suggests widespread expression in the heart, but it is not clear if this was generated using embryonic or adult heart samples as detail on the experimental stage is lacking in the figure legend/panel? The authors need to address this via immunostaining using antibodies against *POFUT1* and markers of each cardiac compartment.
2. Fig. 1b and 1f require accompanying section data to illustrate the gross endothelial-specific *Pofut1* cKO phenotype- in particular for 1f it is unclear why an EC-KO would result in a regionalised infarct (?) and related to this, in the sections in 1g the fibrosis should be quantified with an indication as to whether this is regionalised versus existing throughout the mutant myocardium.
3. Fig. 1j requires quantification to support the conclusion of an increased coronary plexus: specifically vessel branch points, vessel diameter, length etc, should be included. This phenotype resembles more of a remodelling defect, which ought to be accounted for/commented on by the authors.
4. Fig. 2a again supports a remodelling defect, rather than necessarily hypoplasia as stated; there is no evidence that these vessels (in whole mount view) are arteries versus veins. The authors need to investigate a potential venous phenotype (as suggested above), using vein-specific markers such as *Coup-TFII*, *Ephb4* and *Eomucin*. Of note, in zebrafish Notch signalling, mediated by *nr2f1b* controls, venous specification and downstream angiogenic patterning of the developing vasculature (Li et al., 2015).

5. Fig. 2e, f the hypoxyprobe and VEGFA staining are relatively poor quality images and again lack quantification to support the conclusions of a hypoxic zone or expanded VEGFA in the myocardial wall- this data needs improving.
6. Figure 3 is a key figure as relates to the identification of a VEGFR3⁺ coronary “angioblast” population(s), however, the data is based exclusively on immunostaining and conclusions are drawn based on levels of VEGFR3 staining which cannot be accurately determined by this approach. The authors need to study this population in further detail (as noted above) and include lineage-tracing from source into sites of angiogenesis and arteriogenesis within compact myocardium, based on NFATC1-Cre and/or VEGFR3-CreERT2 in combination with VEGFR2, pVEGFR2, pAXILLIN or pERK1/2 staining, as shown in Supplementary Fig. 6c,d for VEGFR3 expression. This is required to validate the hypothesis that in Pofut1cKO hearts the expanded coronary angiogenic population is derived from the endocardium. Low power views (akin to Supplementary Fig. 8) and quantitative analyses (as in Supplementary Fig. 6d) should also be included. Are the microspikes (Fig. 3d, e) enriched for VEGFR3 or DLL4 expression, akin to the endothelial tip cell filopodia-like protrusions? NFATC1-Cre and VEGFR3-CreERT2 based labelling will also facilitate FACS isolation of reporter⁺ cells from hearts at different stages to unequivocally identify VEGFR3^{High} versus VEGFR3^{Low} populations (based on quantitative reporter and VEGFR3 expression) and determine their transcriptional profiles for arterial specification, etc (in both control and Pofut1cKO backgrounds).
7. Figures 3h and 3i require western analyses to quantify pVEGFR2 and pERK11/2 respectively, again this cannot be accurately determined by IMF-alone.
8. The whole mount PECAM staining in Fig. 4a further reinforces the need for the authors to examine the coronary veins in the Pofut1cKO mutants, especially given this analyses focuses on the more surface/superficial vessels, which are generally venous.
9. The images in Fig. 4f are very difficult to interpret around changes in extent of EdU staining- whilst quantitative data is presented in 4g the panels in 4f need improving as they are not currently representative of increased coronary EC proliferation.
10. In Fig. 5, the authors show data to suggest POFUT1 regulates coronary artery formation via NOTCH signalling, but can the phenotype of Pofut1cKO hearts be rescued with gain-of-function of NOTCH signalling, e.g. combination with endocardial-specific NICD overexpression (TGN1CID^{flox/+} mouse)? Also, the authors only examined the requirement for POFUT1 in DLL4 binding to NOTCH1, is this function unique to this NOTCH ligand-receptor pair? This needs to be addressed given that, for example, JAG1-NOTCH3 signalling promotes smooth muscle cell recruitment and vascular maturation and the authors report a defect in pericyte coverage to the extruding endocardium-derived ECs in their explant studies shown in Fig. 2d. Fig. 5f, the authors present no evidence for dedifferentiation of VEGFR3^{Low} to VEGFR3^{High} ECs in the outer myocardial wall, despite the statements on page 9 of the text. This needs toning down and also re-evaluating in light of the suggested NFATC1/VEGFR3-lineage trace experiments (point 6).
11. With regards to the data shown in Fig. 6, what is the functional consequence of deleting

Vegfr3 alone in the endocardium? The IMF panels in Fig. 6 require quantification to support the suggestion of increased coronaries and failed rescue in a VEGFR3cKO background.

12. Similarly, in Fig. 7 the authors characterized the vascular phenotype of Pofut1;Vegfr2 compound mutants, but what is the phenotype of endocardial-Vegfr2 single cKO? Related to this, the information regarding the generation of Vegfr2 cKO mice is missing in the Methods. In addition, the Sirius red panel for the Pofut1cKO heart in Fig. 7d is not indicative of myocardial infarction, in contrast with the panel shown in Fig. 1g. The authors ought to address this point. In addition, the panel set shown in 7i is not convincing with respect to rescue of VEGFR3 and the IMF cannot be reliably used to conclude differences as relates to VEGFR3 High versus Low sub-populations – again this relates to the need for fate mapping and FACS-based molecular phenotyping of these VEGFR3+ populations (point 6).

13. In Supplementary Fig. 5, the characterization of the development of coronary veins and lymphatics vessels in the Pofut1cKO hearts is suboptimal. The authors need to combine immunostaining of sections with whole-heart imaging and include additional vessel (venous, lymphatic) markers, e.g. Endomucin/EPHB4 (coronary veins) and Prox1 (or Podoplanin)/LYVE-1.

14. In Supplementary Fig. 8, the authors need to include low power views of VEGFR2, pVEGFR2, and NICD immunostaining in the Dll4cKO samples.

15. With regards to the data shown in Supplementary Fig. 9, does deletion of Pofut1 and Dll4 (alone or in combination) in this model result in changes in cell motility?

16. In the first sentence of the Discussion the authors claim to “report that VEGFR3^{high} coronary angiogenic precursors undergo vasculogenesis to form the coronary plexuses, which subsequently mature into large coronary arteries by arteriogenesis (Fig. 8)”, this is an overstatement, given the lack of experimental evidences highlighted above, thus the authors need to tone down their conclusions.

17. On page 11, the authors note “Angioblasts were first characterised as vascular tip cells in 2003”; this statement is incorrect and should be amended. Angioblasts are EC precursors that coalesce to form blood vessels de novo, whilst vascular tip cells are a specialized (EC) cell type with high motility, exhibiting filopodia-like protrusions to assist with the expansion of angiogenic vascular beds, i.e. formation of new vessels from pre-existing blood vessels. Of note, endothelial cells displaying filopodia-like protrusions and controlling blood vessel branching morphogenesis were first characterized in 2002 by the Shima lab (Ruhrberg et al Genes & Dev 2002). Later in 2003, these cells were termed vascular tip cells by Gerhardt et al (JCB 2003).

Reviewer #2 (Remarks to the Author):

The authors delete POFUT1, an enzyme required for Notch ligand interaction with receptor, using a variety of Cre lines and demonstrate lethal defects in coronary angiogenesis using

NFATc1 and Tie1 drivers. The defect in coronary vessel growth is demonstrated to be an increase in angiogenic precursor numbers and failure to form more mature arteries. Examination of this phenotype further demonstrates a proliferative, angiogenic endothelial population in the inner myocardium of the normal embryonic heart that is VEGFR3⁺ and expanded following loss of POFUT1. NICD staining confirms a severe loss of Notch signaling in coronary ECs, and timed endothelial deletion of Dll4 in mid-gestation reproduces the coronary arterial defects observed in NFATc1 POFUT1 cKO animals. Most interesting, simultaneous deletion of POFUT1 and VEGFR3 using NFATc1-Cre fails to rescue this phenotype, but simultaneous deletion of POFUT1 and VEGFR2 confers rescue with normal coronary artery development and postnatal heart function. Overall these are well done studies that will be of significant interest to the field. Although some of the concepts (e.g. the role of Notch signaling in negatively regulating VEGFR2 signaling) have been previously shown in the retina and tumor vasculature, these studies provide some interesting new insights into coronary arterial development. These include: (i) the identification of a highly angiogenic inner myocardial population that is tempered by inhibitory Notch signaling in the outer myocardium, (ii) the identification of VEGFR3 as a marker for these angiogenic ECs, and (iii) the demonstration that loss of VEGFR2 and loss of NOTCH signaling compensate for each other and permit normal coronary vessel growth.

Major points

1. If VEGFR3 can compensate for loss of VEGFR2 in R2/POFUT1 DKO animals, why does loss of VEGFR3 fail to rescue loss of NOTCH signaling during coronary vessel development? The authors' studies demonstrate that loss of VEGFR2 is well tolerated in conjunction with loss of NOTCH signaling due to POFUT1 deficiency. They suggest that this is because VEGFR3 functions redundantly with VEGFR2 in angiogenic ECs. However, loss of VEGFR3 fails to rescue loss of NOTCH/POFUT1 function. How can the degree of VEGFR3 function be sufficient to fully support coronary angiogenesis without demonstrating a similar loss of function rescue as VEGFR2? Do the authors believe that the mechanism of NOTCH inhibition is VEGFR2 specific? If so, why and how?
2. What is the role of VEGFR3 in the angiogenic endothelial population of the inner myocardium? The authors do not report the phenotype of NFATc1 VEGFR3 KO mice. Do they form normal coronary vessels? If so, what do the authors believe is the role of VEGFR3 in the angiogenic EC of the normal developing heart? Have the authors performed combinatorial VEGFR2/VEGFR3 NFATc1 KO or POFUT1 cKO rescue studies?

Minor points

1. The Tie1-Cre KO animals add little to the story and can be removed.
2. The figure legends do not adequately describe the genetic studies being shown. Simply

referring to animals as “cKO” is not sufficient, please add specific information to all the panels.

Below are our point-to-point responses (Roman) to the Reviewers' critiques (Italic). Changes are highlighted (blue font) in the revised manuscript and online supplement.

Reviewer #1 (Remarks to the Author):

In this study Wang and colleagues target POFUT1, a modifier of Notch signalling, in the coronary endothelium to reveal a role in angiogenesis and arteriogenesis and identify a putative VEGFR3+ angioblast population, which they suggest may act as a progenitor pool for continuous coronary artery formation. Notch signalling has previously been implicated in coronary artery development, for example Notch 1 has been shown to be important for coronary artery EC commitment and vessel wall maturation (del Monte et al., 2011) and Akt/mTOR acting on Jagged 1 is required for VSMC maintenance such that hearts with loss of Akt and downstream Notch signalling fail to form coronary arteries and arterioles (Kerr et al., 2016). The novelty of the current study, therefore, lies in the targeting of an alternate component of the Notch pathway, POFUT1, and in ascribing function specific to coronary artery formation via misregulated angiogenesis at the level of a putative VEGFR3+ progenitor population residing in the compact myocardium. As such this is an interesting study and one which provides insight into how defective arteriogenesis during development may result in ischaemic heart disease including myocardial infarction and heart failure during adulthood. That said, there are some issues with the study as it currently stands and the data as presented do not fully support the conclusions:

General comments:

The authors have not examined the coronary veins and instead have focused exclusively on

arteries. Some of the analyses presented do not discriminate between veins and arteries for e.g., Fig. 2a and the whole mount PECAM staining in Fig. 4a, and indeed there may well be effects on the more superficial veins according to this data. Notch signalling when impaired is known to lead to arterialisation of veins and consequently an effect on the coronary veins should be investigated and if nothing else excluded from the *Pofut1*^{CKO} phenotype.

More importantly a major proposed finding in this study is the identity of the so-called VEGFR3^{High} sub-population which is suggested to function as a tissue-specific angiogenic population for coronary artery development. The evidence for this is somewhat weak in its current form, with the primary assays based on immunostaining which may simply reflect an up-regulation of VEGFR3 in angiogenic sprouts (as opposed to evidence of a discrete progenitor pool) and which in turn is used (inappropriately) to quantify levels of VEGFR3. Furthermore, the authors defined two angiogenic coronary EC populations based on VEGFR3 expression levels – VEGFR3^{Low} EC in the outer myocardial wall and VEGFR3^{High} EC in the inner myocardium, but it is not clear from their data if both populations are derived from the endocardium or whether these populations represent different states of the same cell type, i.e. angioblast-like cell type expressing high VEGFR3 and angioblast-derived arterial EC expressing low VEGFR3. Likewise, information is lacking as to when this population(s) arises in the developing myocardium, since the authors focused their studies at E16.5. The term “angioblast” may be inappropriately assigned here and requires detailed lineage tracing data, both to reveal the origin of this “angioblast” population and to trace progenitors and their derivatives into developing coronary beds undergoing angiogenesis and/or coalescence into arteries/arterioles within the compact myocardium. Given the potential significance of this finding and that it represents a possible major advance beyond previously published studies, this population needs more detailed characterisation. This should include either an endocardial lineage tracing approach (e.g. based on the expression of *Nfatc1* reporter) combined with immunostaining/flow cytometry or analysis of VEGFR3-CreERT2; reporter fate mapping during coronary vascular development (VEGFR3-CreER2 mice were recently reported by Martinez-Corral et al., 2016), which in addition to labelling *in situ*, would also facilitate isolation of the VEGFR3+ population(s) – High v Low reporter expression and transcriptome analyses to determine artery-specific cell fates etc.

Response: We thank you for reviewing our paper and identifying the novelty of our study. We also thank you for summarizing the function of NOTCH signaling in coronary artery development and have included these key references in the paper. Following your insightful and constructive comments and suggestions, we have substantially revised the paper with new data from experiments. These experiments include the analysis of coronary veins by whole mount and sectional immunostaining for coronary veins and lymphatics using antibodies against Endomucin (EMCN) and LYVE1, respectively. The whole mount staining of EMCN showed that the coronary veins on the surface of the heart were comparable between control and *Pofut1*^{CKO} embryos (Supplementary Fig. 6a). On heart sections, we detected that some intramyocardial arteries in *Pofut1*^{CKO} hearts were positive for venous marker EMCN (Supplementary Fig. 6b). This finding suggests that *Pofut1* might be required for maintaining arterial fate. In addition, we analyzed the cardiac lymphatics by whole mount staining of LYVE1 and found that the lymphatics on the posterior surface of the heart were comparable between control and *Pofut1*^{CKO} embryos, whereas those on the anterior surface of the *Pofut1*^{CKO} heart were reduced

(Supplementary Fig. 7). These findings support that the *Nfatc1*^{Cre}-mediated *Pofut1* deletion primarily affects coronary artery development and suggest that the lymphatic defect (delayed formation) is likely secondary to the coronary artery malformation and poor coronary circulation. The latter is supported by the fact that the *Nfatc1*⁺ lineage doesn't contribute to forming cardiac lymphatics at E16.5 when the lymphatic phenotype has already developed. As recommended, we further investigated the contribution of *Nfatc1*⁺ lineage to coronary arterial, venous and lymphatic endothelium by co-labeling of *Nfatc1*⁺ lineage (GFP) with DLL4, EMCN and LYVE1, respectively. The results showed that the *Nfatc1*⁺ lineage mainly contributed to coronary artery (>70% of total coronary arterial endothelial cells), and had less contribution to coronary vein (<50%), and very little, if any, contribution to cardiac lymphatic endothelium (Supplementary Fig.17-19). These findings are consistent with previous reports by us (Wu *et al.*, *Cell* 2012) and others (Chen *et al.*, *Development* 2014) that *Nfatc1*⁺ ventricular endocardial cells are a major progenitor source of coronary arterial endothelium.

We went on to investigate the origin of VEGFR3^{high} cells by co-staining of VEGFR3 with *Nfatc1*⁺ GFP labeled lineage from E11.5 to E16.5, which covering the entire process of coronary vasculature formation including endocardial sprouting, plexus formation and coalescence, arteriogenesis and maturation. We observed double positive endocardial cells sprouting into coronary sulcus and interventricular septum of E11.5 hearts (Supplementary Fig. 9). Subsequently, the double positive cells forming plexus were observed in the myocardial wall near epicardium, which is relatively more hypoxic than the inner myocardium close to endocardium before coronary circulation begins around E14.5 (Supplementary Fig. 9). By E15.5 with the initiation of coronary circulation, the VEGFR3^{high} cells from the *Nfatc1*⁺ progenitors were predominantly 're-located' in the inner myocardium close to endocardium, corresponding to the shift of hypoxic zone from the outer to inner myocardium (Supplementary Fig. 9, Fig.2e, Fig.3a). Notably, the VEGFR3^{high} cells also expressed high levels of angiogenic marker pVEGFR2 and migratory marker pPAXILLIN. Together, these additional findings support that VEGFR3^{high} cells are actively angiogenic precursors for coronary arteries.

Specific comments:

1. The study lacks a comprehensive analyses of the spatio-temporal expression of Pofut1 in the developing heart to underpin the targeting strategies proposed and to illustrate expression in coronaries (arteries and/or veins). As it stands, the characterization of POFUT1 expression is incomplete; is POFUT1 expressed only in the developing endocardium/endothelium? The data shown in Supplementary Fig. 1a is suboptimal and suggests widespread expression in the heart, but it is not clear if this was generated using embryonic or adult heart samples as detail on the experimental stage is lacking in the figure legend/panel? The authors need to address this via immunostaining using antibodies against POFUT1 and markers of each cardiac compartment.

Response: we apologize for the missing information and poor quality of POFUT1 IF. During revision we tested four commercially available antibodies (ABCAM, ab74302, ab154051; Santa Cruz, sc-98435 and sc-271026) for POFUT1. With sc-271026 antibodies we were able to detect with relatively less background that POFUT1 was predominantly expressed in endocardium and coronary endothelium cells including artery and vein (Supplementary Fig. 1). In *Pofut1*^{CKO} embryos, the expression of POFUT1 in coronary endothelium was greatly reduced. This expression pattern was correlated to the coronary phenotypes resulting from the

endocardial/coronary endothelial deletion of *Pofut1*, but not from its deletion in myocardium or epicardium.

2. *Fig. 1b and 1f require accompanying section data to illustrate the gross endothelial-specific Pofut1cKO phenotype- in particular for 1f it is unclear why an EC-KO would result in a regionalised infarct (?) and related to this, in the sections in 1g the fibrosis should be quantified with an indication as to whether this is regionalised versus existing throughout the mutant myocardium.*

Response: We apologize for the confusion due to inappropriate arrangement of the figures and panels and associated legends. The histological data related to Figure 1b was shown in the Supplementary Fig. 2a-e. We have rearranged the figure panels by moving the old Fig.1f (animal died at P21) to Supplementary Fig. 2f, which describing its histology, and adding a new Fig. 1f (animal died at P34). In this way, the histological data (Fig. 1g) matches new Fig. 1f. Further, we have included quantification for cardiac fibrosis in inner and outer half layer of myocardium, which showing worse fibrosis in the inner myocardium of *Pofut1^{ckO}* hearts (Supplementary Fig. 2c,d). This finding is consistent with the inner myocardium being more hypoxic than outer myocardium in *Pofut1^{ckO}* mice (Fig. 1h).

3. *Fig. 1j requires quantification to support the conclusion of an increased coronary plexus: specifically vessel branch points, vessel diameter, length etc, should be included. This phenotype resembles more of a remodelling defect, which ought to be accounted for/commented on by the authors.*

Response: We tried to quantify the vessel branch points, vessel diameter and length as suggested. It was very difficult to assign these parameters to plexus/vessels, as defective coronaries were so irregular. Alternatively, we quantified the vessel density and the result supports an increase of coronary plexuses in *Pofut1^{ckO}* mice (Supplementary Fig. 2h). Having said that, we agree that *Pofut1^{ckO}* coronaries might have remodeling defects resulting in hypoplastic main coronary arteries and increased coronary arterial networks as shown in Fig. 1i.

4. *Fig. 2a again supports a remodelling defect, rather than necessarily hypoplasia as stated; there is no evidence that these vessels (in whole mount view) are arteries versus veins. The authors need to investigate a potential venous phenotype (as suggested above), using vein-specific markers such as Coup-TFII, Ephb4 and Eomucin. Of note, in zebrafish Notch signalling, mediated by nr2f1b controls, venous specification and downstream angiogenic patterning of the developing vasculature (Li et al., 2015).*

Response: We apologize for insufficient information on the method for Fig. 2a images. These images were taken from a coronary arteriogram using fluorescent dye (Wu *et al.*, *Cell* 2012). In this procedure, fluorescent dye was injected into the left ventricle via the apex of heart. The dye was pumped into coronary arteries through the aorta, thus labeling the coronary arteries, but not the coronary veins. We included a description for obtaining coronary arteriograms in the METHODS. On the other hand, we took your advice and examined coronary veins by EMCN (Endomucin) immunostaining as described above. Whole mount EMCN staining showed that the coronary veins on surface were comparable between control and *Pofut1^{ckO}* embryos

(Supplementary Fig. 6a). On heart sections, we detected that some intramyocardial arteries in *Pofut1^{CKO}* hearts expressed the venous marker EMCN (Supplementary Fig. 6b). Because *Nfatc1* lineage tracing and *Nfatc1*Cre-mediated deletion only reveal an arterial role of POFUT1, we suggest that POFUT1 is required for maintaining arterial fate of endothelium in developing coronaries. However, by no means these findings rule out a potential role of POFUT1 in venous specification. This is one of our future investigations with a venous specific Cre.

5. *Fig. 2e, f the hypoxyprobe and VEGFA staining are relatively poor quality images and again lack quantification to support the conclusions of a hypoxic zone or expanded VEGFA in the myocardial wall- this data needs improving.*

Response: We apologize for the poor quality of images. We now include new images with better qualities in the revised paper (Fig. 2e,g), along with quantifications (Fig. 2f,h). The new data support an expanded hypoxic and VEGFA-expressing zone in the myocardial wall of *Pofut1^{CKO}* embryos (Fig. 2e-h).

6. *Figure 3 is a key figure as relates to the identification of a VEGFR3+ coronary “angioblast” population(s), however, the data is based exclusively on immunostaining and conclusions are drawn based on levels of VEGFR3 staining which cannot be accurately determined by this approach. The authors need to study this population in further detail (as noted above) and include lineage-tracing from source into sites of angiogenesis and arteriogenesis within compact myocardium, based on NFATC1-Cre and/or VEGFR3-CreERT2 in combination with VEGFR2, pVEGFR2, pAXILLIN or pERK1/2 staining, as shown in Supplementary Fig. 6c,d for VEGFR3 expression. This is required to validate the hypothesis that in *Pofut1*CKO hearts the expanded coronary angiogenic population is derived from the endocardium. Low power views (akin to Supplementary Fig. 8) and quantitative analyses (as in Supplementary Fig. 6d) should also be included. Are the microspikes (Fig. 3d, e) enriched for VEGFR3 or DLL4 expression, akin to the endothelial tip cell filopodia-like protrusions? NFATC1-Cre and VEGFR3-CreERT2 based labelling will also facilitate FACS isolation of reporter+ cells from hearts at different stages to unequivocally identify VEGFR3^{High} versus VEGFR3^{Low} populations (based on quantitative reporter and VEGFR3 expression) and determine their transcriptional profiles for arterial specification, etc (in both control and *Pofut1*CKO backgrounds).*

Response: We agree with the reviewer. Accordingly, we further characterized VEGFR3^{high} cells by investigating their origin and angiogenic potential by co-staining VEGFR3 with the *Nfatc1^{Cre}* lineage GFP marker from E11.5 to E16.5. These experiments were performed independently in Zhou and Harvey labs. VEGFR3 was highly expressed in a subset of endocardial cells at the ‘foot’ of trabeculae intimately contacting the compact myocardium at E11.5 when the earliest coronary plexus starts to form in the coronary sulcus and interventricular septum (Supplementary Fig. 9). These VEGFR3^{high} endocardial cells clearly protruded into the myocardium and represented the earliest endocardial ‘sprouts’. VEGFR3^{high} cell population expanded in the myocardium as coronary plexus after E12.5 (Supplementary Fig. 9). At E15.5 (soon after coronary circulation began around E14.5), the fast growing compact myocardium received active perfusion from established coronary arteries located in the outer layer where was thus less hypoxic. In contrast, the inner layer distant from the coronary arteries was more hypoxic (Fig. 2e). These findings suggest the inner myocardium is the region continuously undergoing

vasculogenesis after the initiation of coronary circulation. Consistently, endothelial cells located in the inner myocardium expressed high levels of VEGFR3, while the endothelial cells in matured arteries located in the outer myocardium express low levels of VEGFR3 (Fig. 3a, Supplementary Fig. 9). Importantly, lineage tracing and quantification of VEGFR3^{high} cells at E16.5 shows that majority of VEGFR3^{high} cells was derived from the *Nfatc1*+ endocardial cells (Supplementary Fig. 9). To further determine the temporal change of VEGFR3^{high} cells in the developing coronary arteries, we compared the VEGFR3 expression in the *Nfact1*+ coronary progeny (GFP labeled) between E13.5 and E16.5. The results showed that > 90% GFP expressing cells expressed high levels of VEGFR3 at E13.5, while this number decreased to 60% at E16.5 (Supplementary Fig. 9). This temporal change was well correlated with VEGFR3^{high} cells as angiogenic precursors undergoing vasculogenesis in the forming plexus at E13.5, while reduced VEGFR3 expression in the endothelial cells of mature coronary arteries at E16.5. Moreover, we examined the angiogenic potentials of VEGFR3^{high} cells by co-immunostaining of VEGFR3 with angiogenic markers including VEGFR2, pVEGFR2, pPaxillin and VEGFA. Collectively, the results show that VEGFR3^{high} cells are a subpopulation of VEGFR2 positive cells expressing high levels of pVEGFR2 and pPAXILLIN, and VEGFA levels are high in the inner myocardium where VEGFR3^{high} cells are located (Supplementary Fig. 10). Further, we performed lineage trace for the endocardial derived coronary angioblast precursors and determined the contribution of their progenies to developing coronary arteries, veins and cardiac lymphatics from E11.5 to E16.5 by co-staining tissue-specific markers with the *Nfatc1*Cre-mediated GFP reporter (Supplementary Fig. 17-19). Collectively, these new data support that VEGFR3^{high} cells are angiogenic precursors of coronary arteries derived from the endocardium.

We now include high magnification views of VEGFR3 and DLL4 stained angiogenic precursor cells showing that the microspikes in Fig. 3d also express both angiogenic markers (Fig. 3e for DLL4 and Fig. 3f for VEGFR3). The data support the presence of previously unknown coronary angiogenic precursors. Since our new data from analyzing several angiogenic markers and *Nfatc1*^{Cre}-based lineage trace support that VEGFR3^{high} cells arising from endocardium are angiogenic precursors of coronary arteries, we believe that the transcriptome analysis and *Vegfr3*^{CreERT}-based lineage trace represent a future direction of new investigations to identify new regulators underlying coronary artery formation.

It is worth mentioning that VEGFR3^{high} cells exhibit the main feature of ‘angioblasts’, as coronary plexuses arise *de novo* via vasculogenesis. The term though is ‘traditionally’ used to refer to mesodermal precursors from which vessels arise. On the other hand, coronary arteries arise from endocardium, somewhat resembling ‘angiogenic sprouting’ from an existing endothelial lumen. We, however, opted not to use the term ‘tip cells’ or ‘angioblasts’ for VEGFR3^{high} cells, since neither truly reflects their specific nature and functions in coronary vasculogenesis. In the revised paper, we use the term ‘angiogenic precursors’ for the coronary-forming VEGFR3^{high} cells.

7. Figures 3h and 3i require western analyses to quantify pVEGFR2 and pERK11/2 respectively, again this cannot be accurately determined by IMF-alone.

Response: We have performed western blot analyses as suggested. The results indicate increased

pVEGFR2 and pERK1/2 levels in *Pofut1^{ckO}* hearts (Fig. 3l).

8. The whole mount PECAM staining in Fig. 4a further reinforces the need for the authors to examine the coronary veins in the *Pofut1ckO* mutants, especially given this analyses focuses on the more surface/superficial vessels, which are generally venous.

Response: We agree with the reviewer that whole mount PECAM1 staining mainly labels coronary veins on surface. As discussed above, we have included additional analyses for coronary veins (and cardiac lymphatics). The data are presented in new Supplementary Fig. 6 and 7.

9. The images in Fig. 4f are very difficult to interpret around changes in extent of EdU staining- whilst quantitative data is presented in 4g the panels in 4f need improving as they are not currently representative of increased coronary EC proliferation.

Response: We apologize for the low quality of images. New images with improved quality are included in new Supplementary Fig. 11f (previous Fig. 4f). We re-organized Figures by putting Fig. 4 into Supplement as Supplementary Fig. 11 per Reviewer 2's recommendation.

10. In Fig. 5, the authors show data to suggest *POFUT1* regulates coronary artery formation via *NOTCH* signalling, but can the phenotype of *Pofut1ckO* hearts be rescued with gain-of-function of *NOTCH* signalling, e.g. combination with endocardial-specific *NICID* overexpression (*TGNICID^{flox/+}* mouse)? Also, the authors only examined the requirement for *POFUT1* in *DLL4* binding to *NOTCH1*, is this function unique to this *NOTCH* ligand-receptor pair? This needs to be addressed given that, for example, *JAG1-NOTCH3* signalling promotes smooth muscle cell recruitment and vascular maturation and the authors report a defect in pericyte coverage to the extruding endocardium-derived ECs in their explant studies shown in Fig. 2d. Fig. 5f, the authors present no evidence for dedifferentiation of *VEGFR3^{Low}* to *VEGFR3^{High}* ECs in the outer myocardial wall, despite the statements on page 9 of the text. This needs toning down and also re-evaluating in light of the suggested *NFATC1/VEGFR3*-lineage trace experiments (point 6).

Response: We have previously generated mice with overexpression of N1ICD in endocardium using *Nfatc1^{Cre}* and the mutant embryos died before E11.5 (please see Figure below). The early lethality prevents us to test whether overexpressing N1ICD could rescue

coronary defects in *Pofut1^{ckO}* embryos. *POFUT1* can modify all *NOTCH* receptors and affect binding by all ligands (Stahl *et al.* *JBC* 2008). In this study, we focused on *DLL4* binding to *NOTCH1* because they are the major ligand and receptor expressed in endocardium and endothelium. Loss of *Pofut1* does reduce binding of *DLL4*. This is consistent with the previous report that *FRINGE*

Overexpression of N1ICD in endocardium using *Nfatc1^{Cre}* results in the death of embryos at E11.5-12.5.

modification of NOTCH receptors, which acting downstream of POFUT1, favors DLL4 binding (Benedito *et al.*, *Cell* 2009; Amato *et al.*, *Nature cell biology* 2015).

We thank the reviewer for pointing out the role of JAG1-NOTCH3 signaling in regulating smooth muscle cell recruitment and differentiation. JAG1 in the endothelium signals to NOTCH3 in smooth muscle cells for vessel maturation (Domenga *et al.*, *Genes Dev* 2004; High *et al.*, *PNAS* 2008; Liu *et al.*, *Circ Res* 2009). In the present study, *Pofut1* was deleted in coronary endothelium, which is unlikely to have a direct effect on the function of NOTCH3 expressed in smooth muscle cells, unless *Pofut1* deletion reduced JAG1 expression and/or function. Our qPCR results indicated reduced *Jag1* mRNA levels in *Pofut1*^{CKO} hearts. We speculate that such reduced *Jag1* expression may result in less JAG1/NOTCH3 signaling between coronary endothelium and smooth muscle cells, thus affecting smooth muscle cell recruitment and vascular maturation, which we considered as a secondary effect, as POFUT1 modifies NOTCH receptors, not ligands.

The additional data discussed above support a unique population of VEGFR3^{High} angiogenic precursors arising from the endocardium and a spatiotemporal shift from VEGFR3^{High} expression in active proliferating plexus to VEGFR3^{Low} expression in mature arteries. Related to the *Dll4* deletion experiments (previous Fig. 5f), we quantified the percentages of VEGFR3^{High} and VEGFR3^{Low} cells. The results showed that the percentage of VEGFR3^{Low} cells in *Dll4*^{CKO} (3.7%) was much less than it in control animals (53.4%), while the percentage of VEGFR3^{High} cells in *Dll4*^{CKO} (96.2%) was much more than it in control (46.5%) (Fig. 4f). These findings indicate that loss of *Dll4* promotes the VEGFR3^{Low} cells to become VEGFR3^{High} cells. Even with these new findings, we still consider a need for additional markers and functional characteristics to claim ‘dedifferentiation’ of VEGFR3^{High} angiogenic precursors induced by *Dll4* deletion. We thus have removed the term from the paper.

11. With regards to the data shown in Fig. 6, what is the functional consequence of deleting Vegfr3 alone in the endocardium? The IMF panels in Fig. 6 require quantification to support the suggestion of increased coronaries and failed rescue in a VEGFR3cKO background.

Response: The data for endocardial deletion of *Vegfr3* are included, which show that the deletion promotes coronary plexus formation (Supplementary Fig. 16a). This finding is consistent with previous observations in mouse retina angiogenesis model (Tammela *et al.*, *Nat Cell Biol.* 2011). Quantification of vessel density for the IMF panels in Figure 6 is also included. The results show that deletion of *Vegfr3* fail to rescue the coronary phenotypes in *Pofut1*^{CKO} or *Dll4*^{CKO} embryos (Supplementary Fig. 16d,e).

12. Similarly, in Fig. 7 the authors characterized the vascular phenotype of Pofut1;Vegfr2 compound mutants, but what is the phenotype of endocardial-Vegfr2 single cKO? Related to this, the information regarding the generation of Vegfr2 cKO mice is missing in the Methods. In addition, the Sirius red panel for the Pofut1cKO heart in Fig. 7d is not indicative of myocardial infarction, in contrast with the panel shown in Fig. 1g. The authors ought to address this point. In addition, the panel set shown in 7i is not convincing with respect to rescue of VEGFR3 and the IMF cannot be reliably used to conclude differences as relates to VEGFR3 High versus Low sub-populations – again this relates to the need for fate mapping and FACS-based molecular

phenotyping of these VEGFR3+ populations (point 6).

Response: We now include the data for endocardial deletion of *Vegfr2*. *Vegfr2* deletion represses coronary angiogenesis (Supplementary Fig. 16b). Detailed information on generation of *Vegfr2*^{CKO} is also included in Supplementary Materials and Methods. Figure 7d represents Sirius Red staining of heart sections from *Pofut1*^{CKO} mice of two-month age. They represent a group of *Pofut1*^{CKO} mice without the early onset of acute severe myocardial infarction occurring at 3 week of age (also in Fig. 1). In contrast, they develop heart failure by 2 months of age with increased cardiac fibrosis (Fig. 6d). This increased fibrosis was abolished in mice with double deletion of *Pofut1* and *Vegfr2* (Fig. 6d,e). The Fig. 1g images are the Sirius Red staining of heart sections from mice died at ~one-month old due to acute myocardial infarction. We have further characterized the VEGFR3^{high} cells in response to the point 6 as discussed above.

13. In Supplementary Fig. 5, the characterization of the development of coronary veins and lymphatics vessels in the Pofut1cKO hearts is suboptimal. The authors need to combine immunostaining of sections with whole-heart imaging and include additional vessel (venous, lymphatic) markers, e.g. Endomucin/EPHB4 (coronary veins) and Prox1 (or Podoplanin)/LYVE-1.

Response: As mentioned above, we have addressed these weaknesses by performing additional experiments per your recommendations. The whole mount staining of EMCN shows that coronary veins on heart surface are comparable between control and *Pofut1*^{CKO} embryos (Supplementary Fig. 6a). On heart sections, however, some intramyocardial arteries in *Pofut1*^{CKO} hearts begin to express venous marker EMCN (Supplementary Fig. 6b), suggesting *Pofut1* is required for maintaining arterial fate. For cardiac lymphatics, whole mount staining of LYVE1 shows while the lymphatics on the posterior surface of the heart are comparable between control and *Pofut1*^{CKO} embryos, those on the anterior surface of *Pofut1*^{CKO} heart were underdeveloped (Supplementary Fig. 7). These findings support that the *Nfatc1*^{Cre}-mediated *Pofut1* deletion primarily affects coronary artery development and suggest that the delayed lymphatic formation is likely secondary to coronary artery malformation and poor coronary circulation. The latter is supported by the fact that the *Nfatc1*+ lineage doesn't contribute to forming cardiac lymphatics at E16.5 (Supplementary Fig. 19), when the lymphatic phenotype has already developed.

14. In Supplementary Fig. 8, the authors need to include low power views of VEGFR2, pVEGFR2, and NICD immunostaining in the Dll4cKO samples.

Response: We now include the low power views of VEGFR2, pVEGFR2 and NICD immunostaining in the *Dll4*^{CKO} hearts (Supplementary Fig.14).

15. With regards to the data shown in Supplementary Fig. 9, does deletion of Pofut1 and Dll4 (alone or in combination) in this model result in changes in cell motility?

Response: We have performed these experiments accordingly. The results indicate that knockdown of *Pofut1* or *Dll4* with shRNA promotes cell migration (Supplementary Fig. 15a,c,d).

16. *In the first sentence of the Discussion the authors claim to “report that VEGFR3^{high} coronary angiogenic precursors undergo vasculogenesis to form the coronary plexuses, which subsequently mature into large coronary arteries by arteriogenesis (Fig. 8)”, this is an overstatement, given the lack of experimental evidences highlighted above, thus the authors need to tone down their conclusions.*

Response: In light of the new data which support the presence of the VEGFR3^{high} cells expressing angiogenic markers in the active forming coronary plexus, we suggest that these cells generate early coronary vessels by vasculogenesis as a working model (Figure 7). We also speculate the defective arteriogenesis as secondary to the early plexus defect. Given the speculative nature, we have taken the reviewer’s suggestion and toned down the conclusions.

17. *On page 11, the authors note “Angioblasts were first characterised as vascular tip cells in 2003”; this statement is incorrect and should be amended. Angioblasts are EC precursors that coalesce to form blood vessels de novo, whilst vascular tip cells are a specialized (EC) cell type with high motility, exhibiting filopodia-like protrusions to assist with the expansion of angiogenic vascular beds, i.e. formation of new vessels from pre-existing blood vessels. Of note, endothelial cells displaying filopodia-like protrusions and controlling blood vessel branching morphogenesis were first characterized in 2002 by the Shima lab (Ruhrberg et al Genes & Dev 2002). Later in 2003, these cells were termed vascular tip cells by Gerhardt et al (JCB 2003).*

Response: We apologize for missing the original work defining the ‘tip cells’ and include this important paper in references. As discussed above, we are aware of the distinct characteristics between de novo vasculogenesis and angiogenesis from pre-existing vessels. Given that coronary vessels arise *de novo* from vasculogenesis, we use the term ‘vasculogenesis’ in the model (Fig. 7). On the other hand, ‘VEGFR3^{high}’ cells possess features of angiogenic precursors and sprout from the endocardial sheet, which resemble the initial vascular branching, yet they don’t form new lumens connecting to endocardium. Therefore, early coronary plexus formation consists of both vasculogenic and angiogenic features, and represents a unique organ specific new vessel formation. Such unique features cause confusions and certainly problems when describing coronary formation by the VEGFR3^{high} cells. When characterizing VEGFR3^{high} cells and their involvement in coronary artery formation in the paper and the model, we use a descriptive term ‘angiogenic precursors’ for their sprouting feature analogue to angiogenic cells and precursor function in vasculogenesis.

Reviewer #2 (Remarks to the Author):

The authors delete POFUT1, an enzyme required for Notch ligand interaction with receptor, using a variety of Cre lines and demonstrate lethal defects in coronary angiogenesis using NFATc1 and Tie1 drivers. The defect in coronary vessel growth is demonstrated to be an increase in angiogenic precursor numbers and failure to form more mature arteries. Examination of this phenotype further demonstrates a proliferative, angiogenic endothelial population in the inner myocardium of the normal embryonic heart that is VEGFR3⁺ and expanded following loss of POFUT1. NICD staining confirms a severe loss of Notch signaling in coronary ECs, and timed endothelial deletion of Dll4 in mid-gestation reproduces the coronary arterial defects observed in NFATc1 POFUT1 cKO animals. Most interesting, simultaneous

deletion of POFUT1 and VEGFR3 using NFATc1-Cre fails to rescue this phenotype, but simultaneous deletion of POFUT1 and VEGFR2 confers rescue with normal coronary artery development and postnatal heart function. Overall these are well done studies that will be of significant interest to the field. Although some of the concepts (e.g. the role of Notch signaling in negatively regulating VEGFR2 signaling) have been previously shown in the retina and tumor vasculature, these studies provide some interesting new insights into coronary arterial development. These include: (i) the identification of a highly angiogenic inner myocardial population that is tempered by inhibitory Notch signaling in the outer myocardium, (ii) the identification of VEGFR3 as a marker for these angiogenic ECs, and (iii) the demonstration that loss of VEGFR2 and loss of NOTCH signaling compensate for each other and permit normal coronary vessel growth.

Major points

1. If VEGFR3 can compensate for loss of VEGFR2 in R2/POFUT1 DKO animals, why does loss of VEGFR3 fail to rescue loss of NOTCH signaling during coronary vessel development? The authors' studies demonstrate that loss of VEGFR2 is well tolerated in conjunction with loss of NOTCH signaling due to POFUT1 deficiency. They suggest that this is because VEGFR3 functions redundantly with VEGFR2 in angiogenic ECs. However, loss of VEGFR3 fails to rescue loss of NOTCH/POFUT1 function. How can the degree of VEGFR3 function be sufficient to fully support coronary angiogenesis without demonstrating a similar loss of function rescue as VEGFR2? Do the authors believe that the mechanism of NOTCH inhibition is VEGFR2 specific? If so, why and how?

Response: We thank the reviewer for identifying the significance of our study. Our new data show that deletion of VEGFR3 promotes coronary angiogenesis (Supplemental Fig. 16a); therefore VEGFR3 likely negatively regulates coronary plexus formation. Previous studies have shown that VEGFR3 cannot compensate for loss of *Vegfr2* in the context of over-angiogenesis caused by NOTCH inhibition in mouse retina (Zarkada *et al.* PNAS 2015). Similarly, our data indicate VEGFR3 cannot compensate for loss of *Vegfr2* in *Vegfr2/Pouft1* hearts in coronary plexus formation. On the other hand, the restored coronary plexus formation by VEGFR2 deletion in the *Vegfr2/Pouft1* DKO hearts might be due to incomplete deletion of *Vegfr2* (data not shown), and loss of *Pofut1* activates VEGFR2 in the 'escaped' cells and promotes their proliferation to supplement those *Vegfr2*-null cells for coronary plexus formation. Our data show that deletion of *Pofut1* or *Dll4* promotes both VEGFR2 and VEGFR3 expression, suggesting that the mechanisms of NOTCH inhibition are not VEGFR2 specific.

2. What is the role of VEGFR3 in the angiogenic endothelial population of the inner myocardium? The authors do not report the phenotype of NFATc1 VEGFR3 KO mice. Do they form normal coronary vessels? If so, what do the authors believe is the role of VEGFR3 in the angiogenic EC of the normal developing heart? Have the authors performed combinatorial VEGFR2/VEGFR3 NFATc1 KO or POFUT1 cKO rescue studies?

Response: We have included the data from deletion of *Vegfr3* using the *Nfatc1^{Cre}* driver. The results show that deletion of *Vegfr3* results in increased coronary vessels (Supplemental Fig. 16a). This finding is consistent with previous observations in mouse retina models (Tammela *et*

al. Nat Cell Biol. 2011). This also explains why *Vegfr3* deletion fails to rescue the over-angiogenic defect resulting from deletion of *Pofut1* or *Dll4*. Previous studies in mouse retina suggest that VEGFR2 is required for hypersprouting resulting from *Vegfr3* deletion or NOTCH inhibition of (Zarkada *et al. PNAS* 2015), supporting a dominant role of VEGFR2 in promoting retina angiogenesis. Similarly, our rescue experiments also support such a function for VEGFR2. We agree double deletion of *Vegfr2/Vegfr3* in the endocardial/coronary endothelial lineage might reveal discrete or collaborative functions of VEGFR2 and VEGFR3, and these experiments will be a direction of future studies.

Minor points

1. *The Tie1-Cre KO animals add little to the story and can be removed.*

Response: We have moved the Tie1-Cre KO data as Supplementary Fig. 11.

2. *The figure legends do not adequately describe the genetic studies being shown. Simply referring to animals as “cKO” is not sufficient, please add specific information to all the panels.*

Response: We apologize for using the ‘unconventional’ labeling for genetic studies. This is due to lack of space for captions in Figures. Since we have moved the Tie1-Cre data to Supplementary figures, the *Pofut1^{cKO}* is specifically representing *Pofut1* KO mediated by *Nfatc1^{cre}* driver. In the revised paper, we also include full names for abbreviations in Figure legends as well as in the main text.

Reviewers' Comments:

Reviewer #1 (Remarks to the Author):

The first version of the manuscript had several issues which needed to be addressed experimentally and the authors have done a commendable job in tackling these head-on and improving the study significantly with the inclusion of new data.

Most notably the characterisation of the coronary veins (Supplementary Fig. 6a) and lymphatics (Supplementary Fig. 7) in the *pofut1cKO* mutants, the contribution of the *Nfatc1+* lineage to arterial versus venous ECs (Supp. Figs 17-19) and the more in depth spatial characterisation of the VEGFR3-high cell contribution to the developing coronaries in the context of the *Nfatc1+* lineage (Supplementary Fig. 9, Fig. 2e, Fig. 3a), are all welcome additions which improve the study as a whole.

There are no outstanding issues following the revisions and the major finding is now conclusive in attributing VEGFR3^{high} cells as angiogenic precursors for the coronary arteries. As such this study is significant in offering novel and interesting insights into coronary vessel development.

Reviewer #2 (Remarks to the Author):

The authors have responded very thoroughly to the comments of both reviewers. I will not address whether or not VEGFR3⁺ cells meet the standard for "angioblasts", but the rest of the manuscript is very complete and well done. I particularly appreciate Supp. Fig. 16 that more fully describes the role of VEGFR3 in the developing coronary vasculature.

Reviewer #1:

The first version of the manuscript had several issues which needed to be addressed experimentally and the authors have done a commendable job in tackling these head-on and improving the study significantly with the inclusion of new data.

*Most notably the characterisation of the coronary veins (Supplementary Fig. 6a) and lymphatics (Supplementary Fig. 7) in the *pofut1cKO* mutants, the contribution of the *Nfatc1+* lineage to arterial versus venous ECs (Supp. Figs 17-19) and the more in depth spatial characterisation of the VEGFR3-high cell contribution to the developing coronaries in the context of the *Nfatc1+* lineage (Supplementary Fig. 9, Fig.2e, Fig.3a), are all welcome additions which improve the study as a whole.*

There are no outstanding issues following the revisions and the major finding is now conclusive in attributing VEGFR3high cells as angiogenic precursors for the coronary arteries. As such this study is significant in offering novel and interesting insights into coronary vessel development.

Response: We thank you again for reviewing our paper and appreciate your encouraging comments on our work.

Reviewer #2:

The authors have responded very thoroughly to the comments of both reviewers. I will not address whether or not VEGFR3+ cells meet the standard for "angioblasts", but the rest of the manuscript is very complete and well done. I particularly appreciate Supp. Fig. 16 that more fully describes the role of VEGFR3 in the developing coronary vasculature.

Response: We thank you again for reviewing our paper and identifying the significance of our new data. Following your suggestion, we have replaced the term "angioblast" with "angiogenic precursor" throughout the paper.